# Spiritual Connectivity Intervention for Individuals with Depressive Symptoms: A Randomized Control Trial

**DOI:** 10.3390/healthcare12161604

**Published:** 2024-08-12

**Authors:** Judy Leung, Kin-Kit Li

**Affiliations:** 1School of Nursing & Health Studies, Hong Kong Metropolitan University, Hong Kong SAR, China; 2Department of Social and Behavioural Sciences, City University of Hong Kong, Hong Kong SAR, China

**Keywords:** spiritual intervention, faith, depression, connectivity, Christian, community

## Abstract

Depression is one of the most prevalent mental disorders worldwide. This study examined the effect of a spiritual connectivity intervention on individuals with depression in a randomized waitlist-controlled trial. Fifty-seven participants with mild or moderate depressive symptoms were randomly assigned to either the intervention group (*n* = 28) or the waitlist control group (*n* = 29). The intervention comprised eight weekly sessions focusing on divine connection, forgiveness and freedom, suffering and transcendence, hope, gratitude, and relapse prevention. The outcome measures included depressive symptoms, anxiety, hope, meaning in life, self-esteem, and social support. Participants completed self-administered questionnaires at baseline (week 0), post-intervention (week 8), and 3-month follow-up (week 20). Repeated-measures ANOVA and one-way ANCOVA were used to compare the within-group and between-group differences in the changes in outcome variables. Participants in the intervention group showed significant improvements in depression, anxiety, spiritual experience, hope, self-esteem, and perceived social support after the intervention. Effect size statistics showed small to large differences (Cohen’s *d*, 0.308 to −1.452). Moreover, 85.71% of participants in the intervention group also experienced clinically significant reductions in PHQ-9 scores from baseline to immediate post-intervention. This study highlights the effectiveness of a low-cost, accessible intervention suitable for community implementation by clergy and faith-based organizations.

## 1. Introduction

Depression is a prevalent and severe mental health issue that affects around 280 million people globally [1]. It is the top contributor to global disability and carries the risk of self-harm and suicide, making it one of the leading causes of mortality [2].

While numerous mental health interventions exist, many individuals still struggle to access effective treatment. Depression is one of the most under-reported and under-treated mental health issues [3]. This may be due to ignorance or a lack of knowledge about their health problem. People’s belief in mental health or stigmatization might also hinder their help-seeking behavior [4]. Current approaches often fall short in addressing the complex needs of those with mental health disorders [5]. There is an urgent need to explore novel, evidence-based interventions that can be integrated into existing services, improving accessibility and enhancing outcomes [6,7].

An increasing number of patients with depression have sought non-pharmacological alternatives owing to the drawbacks of antidepressants [8]. Complementary therapies, including religious/spiritual interventions, are increasingly sought by individuals with depression [9]. This study developed and examined the effects of a spiritual connectivity intervention (SCI) as a group community program for those with depressive symptoms, emphasizing Christian beliefs owing to its widespread influence globally [10,11]. What distinguishes our study is the explicit focus on fostering connectedness, a vital component that sets it apart from other interventions in this domain. By implementing a community-oriented approach, we aimed to bridge the existing service gap and extend support to a segment of the population that is often hard to reach. This initiative is particularly significant as it aims to facilitate early identification and intervention for individuals who might otherwise lack access to or engagement with conventional mental health services, potentially alleviating the strain on the public healthcare system.

### 1.1. Spirituality, Religion, and Depression

Depression encompasses a complex interplay of connections and disconnections within the journey of life [12]. It can lead to disconnections with oneself, others, and society, resulting in spiritual disconnection, a lack of personal direction, loneliness, and spiritual crises [13,14]. Addressing these disconnections is crucial, as they can contribute to various physical and emotional problems [15]. Over the past few decades, mental health professionals have displayed an increasing interest in incorporating psychosocial and spiritual care for mental health recovery [16,17]. While studies generally suggest positive associations between religion/spirituality and depression, negative effects can also arise. Individuals may experience guilt and shame when they fail to meet their religious/spiritual community’s behavioral expectations, which can lead to anxiety and depression [18]. Engaging in negative religious coping mechanisms, such as blaming God for personal hardships, has been linked to increased depressive symptoms [19]. Healthcare professionals need to be sensitive to clients’ spiritual needs as they can influence the progress of their illness.

There is a growing interest in exploring the relationships between religion, spirituality, and mental health as integral components of holistic care. Holistic care acknowledges that individuals with mental health conditions have interconnected psychological, social, physical, and spiritual needs [20]. Portnoff et al. highlighted that heightened spirituality correlated with reduced depression risk across diverse cultural settings, yet the heavy reliance on self-reported data raises concerns about potential biases [21]. Anderson et al. demonstrated the efficacy of faith-adapted cognitive behavior therapy (CBT) in treating depression, outperforming standard CBT and control conditions with a moderate effect size ranging from 0.31 to 0.59 [22]. However, the lack of detailed randomization descriptions and researcher allegiance in these trials introduces potential biases. Similarly, Captari et al. showcased the benefits of R/S-adapted psychotherapy over standard treatment for depression, with an effect size of Hedges’ g = 0.33 [23]. Nonetheless, they noted the absence of proper comparative secular and religious/spiritual treatments in many studies, impacting result interpretation.

Clergy members have long provided counseling and care to church members, with a higher percentage of individuals seeking help for their mental disorders from clergy (25%) compared with psychiatrists (16.7%) and general medical doctors (16.7%) [24]. Implementing a community program via clergy members or in churches can be a practical approach that reaches individuals with depressive symptoms who may not seek any treatment otherwise. Clergy should refer the individuals to mental health services if their condition deteriorates or becomes unmanageable.

### 1.2. Therapeutic Components

Research has consistently shown that higher religious engagement is associated with fewer depressive symptoms and quicker recovery from depressive disorders [19,25,26]. Previous studies have demonstrated a positive relationship between religion/spirituality and enhanced mental health [27,28,29,30,31]. Promising results have been observed in faith-based interventions, suggesting their potential practical application [22,32,33,34,35].

Spirituality encompasses the individual’s profound sense of connection with everything other than the self [36]. It involves a deep connection that extends beyond individual existence, encompassing a broader, transcendent dimension [37,38]. Spiritual connectedness includes both a vertical magnitude, representing the relationship with the transcendent/God, and a horizontal magnitude, reflecting connections with others, and, hence, it is influenced by shifts in values, beliefs, and interpersonal interactions, allowing for a transformative experience [39,40]. Spirituality and religion can serve as catalysts for individuals to discover meaning in their suffering, transcending the experience of pain and distress [41].

Extensive research has consistently demonstrated that forgiveness significantly reduces depressive symptoms [42,43]. It is essential that forgiveness should be approached in a personalized manner, taking into account the unique circumstances, beliefs, and experiences of individuals dealing with depression [44,45]. Forgiveness includes not only forgiving others but also the practice of forgiving oneself, allowing individuals to free themselves from a victim mindset and experience genuine freedom [46]. Forgiveness contributes to personal growth and spiritual development and is positively associated with spirituality [47,48]. Moreover, forgiveness benefits not only one’s well-being but also improves relationships and connections with others [49]. True freedom can be maintained through forgiveness [50], thereby enhancing connectivity to the self, interpersonal connectivity, and spiritual connectivity.

Hope has been consistently found to be inversely associated with depression in numerous studies [51,52,53]. Healthcare providers can support patients by facilitating their exploration of life’s meaning and the pursuit of life goals and teaching them skills to regulate their emotions and adopt healthy lifestyles [54].

Gratitude is inversely related to depression [55,56,57] and can enhance spirituality [58]. When individuals express gratitude, they develop a positive view, leading to a stronger connection to the world and enhanced life experiences, which can lead to spiritual growth [59].

The literature reveals notable gaps in the knowledge concerning spiritual interventions for individuals with depression. Firstly, there is a dearth of studies using rigorous methodological designs, such as randomized controlled trials (RCTs), to evaluate the efficacy of spiritual interventions for individuals with depression. Secondly, there is a scarcity of research investigating the application of spiritual interventions for depression within the Chinese context.

### 1.3. Current Study

To address these gaps, an SCI was developed to offer participants information on spiritual coping strategies that could be integrated into their daily lives. The concept of connectivity was the main theme in one weekly session and was incorporated into other sessions and components of the SCI. The recovery process was characterized as a transformative journey of self-discovery and self-renewal, involving the adjustment of attitudes, feelings, perceptions, and beliefs about oneself, others, and life [14]. Recognizing the close connection between spiritual renewal and recovery, this program’s rationale aligned with the specific needs of the target population, aiming to facilitate their spiritual growth and overall well-being. We hypothesized that a community-oriented SCI that focuses on connectedness can reduce depressive symptoms and anxiety levels and enhance hope, meaning in life, self-esteem, and social support. In addition, we examined the moderation effect of demographic variables on the intervention’s effects.

## 2. Materials and Methods

### 2.1. Study Design

This study was a single-blinded, two-arm randomized control trial with a 3-month follow-up. Participants were randomized into either the SCI group (SCG) or the waitlist control group (WLG). A waitlist design was chosen because of the absence of a suitable sham intervention program; this design is commonly used in the evaluation of psychotherapy for depression [60]. This trial was registered with ClinicalTrial.gov (NCT04631900) and its reporting adhered to the guidelines of the CONSORT 2010 Statement [61].

### 2.2. Setting and Participants

The participants were recruited from non-government organizations, local churches, tertiary institutions, district councils of Hong Kong, personal networks, and snowballing. The inclusion criteria were as follows: (i) Hong Kong Chinese residents who can communicate in Cantonese, (ii) aged 18–64, (iii) no objection to Christian rituals, (iv) scored between 5 and 14 out of 27 on the Patient Health Questionnaire-9 (PHQ-9; see further details below), and (v) willingness to comply with the trial’s protocol. Details of the PHQ-9 can be found in the section on outcome measures. The exclusion criteria were as follows: (i) those who had received any form of psychotherapy in the past 3 months, (ii) those with significant cognitive impairment, (iii) those with a lifetime history of psychosis that would make them unable to understand or follow instructions, (iv) those with a strong suicidal risk, or (v) those who had adjusted their medication (antidepressants) within the past 3 months.

### 2.3. Sample Size

A meta-analysis on faith-based intervention reported a small-to-medium effect size of 0.31–0.59 [22]. Another meta-analysis of psychotherapy for adult depression revealed that the effects for high-quality studies (d = 0.22) were significantly smaller than those for other studies (d = 0.74) [62]. Using G*Power version 3.1.9.7 statistical software [63], considering a conservative effect size of 0.38 (f = 0.19), an alpha level of 0.05, and a statistical power of 0.8 [64], with repeated-measures ANOVA tests used for comparisons within and between two groups, resulted in a required sample of 58.

### 2.4. Recruitment and Screening for Participants

The recruitment spanned August 2021 to October 2022. Those who were interested completed an online questionnaire (PHQ-9). Then, they were first screened in a telephone interview to determine whether preliminary inclusion/exclusion criteria were met. Subsequently, they attended a psychiatric interview conducted by one of the investigators, an experienced psychiatric nurse, to rule out cognitive impairment or strong suicidal thoughts. Any participants who exhibited these concerns were advised to seek professional advice. Two intended participants were referred to psychiatric consultation and a non-governmental organization.

### 2.5. Randomization, Allocation, and Blinding

An online, computerized sequence generation randomization tool (https://www.sealedenvelope.com/, accessed on 16 November 2021) was used. A list of block sequences of two group labels (A = intervention group, B = control group) was generated by an independent administrator who was not involved in this study. After that, the group labels (A or B) were marked on pieces of paper and placed inside sequentially numbered, opaque sealed envelopes. The allocation lists were concealed and kept by the independent administrator. Given the nature of the psychological intervention, it was not feasible to blind the research investigator. All the participants were blinded to the allocation.

### 2.6. Ethical Considerations

This study was approved by the Human Subjects Ethics Sub-Committee of the City University of Hong Kong (ref. no.: 14-2020-16E) and the Hong Kong Metropolitan University (formally named Open University of Hong Kong) (ref. no.: HE-SF2021/08). Approval from the respective Boards of Directors of the NGOs, tertiary institutions, and local churches was sought. Written consent was obtained from the participants before randomization.

### 2.7. Interventions

The program drew upon a theoretical framework that integrated elements of spirituality/religiosity within a group therapy approach, incorporating principles from cognitive behavioral therapy and positive psychology intervention (Figure 1) [35].

To ensure the effectiveness, clinical suitability, and practicality of implementing the program, the protocol underwent a rigorous evaluation process. It was reviewed by a panel of experts, comprising a psychiatrist, psychiatric nurses, social workers, and academic scholars, who provided their valuable insights and feedback. Based on the comments and suggestions from these experts, the protocol was revised and modified.

The intervention integrated cognitive reframing and behavioral activation, which are core skills in cognitive behavioral therapy [65]. This approach emphasizes rationality and avoiding negative and irrational thinking that can impact emotions and behaviors. Scriptural passages were used both in meditative or prayerful recitation (behavior) and as a means of shifting one’s perspective (cognition).

Positive psychology intervention strategies and skills were employed to foster optimism, appreciation of the present, acceptance of the past, gratitude, forgiveness, and a broader perspective on life beyond immediate joys and sorrows. Research suggests that positive psychology interventions can cultivate positive emotions, behaviors, and cognitions; enhance well-being; and alleviate depressive symptoms [66].

In a group setting, participants experienced therapeutic elements such as altruism, universality, hope, cohesiveness, and catharsis [67]. Peer support within the group enhanced the participants’ recovery journey. The SCI intervention framework was grounded in Christianity, incorporating Bible verses, prayer, hymn singing, and mutual support among group members. Participants came together in a state of spiritual communion and unity, experiencing personal, social, and spiritual connection and the transformative power of healing.

The SCI was conducted every week for eight weeks. The sessions addressed various topics related to spirituality, connectivity, and the enhancement of mental health. The integration of spirituality/religiosity in the group therapy approach alongside cognitive behavioral therapy and positive psychology principles was implemented in every session (Appendix A). In response to the feedback received during the pilot study, the program was extended from 6 sessions to 8 sessions with the same content to allow additional time for the participants to share and interact [35]. Each session consisted of three parts. The first part lasted approximately 15 min and involved activities such as singing hymns, reviewing homework, and facilitating the participants’ sharing of their applications of the previous week’s topic. The second part, lasting around 90 min with a 10 min break, focused on engaging the participants in activities and discussions related to the specific theme of the week. Ample time was allocated for processing, discussion, and practical exercises. The final part lasted for approximately 15 min and served as a wrap-up for the session and an introduction to assignment tasks. Additionally, to foster hope and establish a spiritual connection, each session was concluded with a prayer. While the majority of the participants were Christians, their faith helped them see a common bond among all people, regardless of whether they were religious or not; social connectedness was also enhanced when they prayed for other people [68]. The participants learned to be with themselves through personal quiet time to better handle solitude and discover inner feelings and thoughts [14]. They were encouraged to enhance self-care, personal acceptance, personal values, and personal feelings as strategies to promote connectedness to oneself.

A mobile app was developed to aid the intervention delivery. Communication with participants could be enhanced by posting announcements via the app, allowing them to communicate in a chat room. It also provided a diary for them to record their daily spiritual activities, including daily quiet time, Bible reading, meditation/reflection, prayer, hymn singing, thanksgiving, and fellowship. Furthermore, the app also provided daily scripture and YouTube videos on hymns, prayers, and spiritual meditation.

### 2.8. Spirituality Connectivity Intervention Group (SCG)

During the intervention period, the participants in the SCG were asked not to start any pharmacotherapeutic, herbal, or psychotherapeutic treatments or change their usual drug treatments for depressive symptoms, and adherence to the intervention sessions was monitored. The initial 8-week program phase was followed by a 3-month (20-week) follow-up period with no restrictions on other treatments for depression.

### 2.9. Waitlist Control Group (WLG)

Having completed the baseline measure, the WLG did not receive any form of intervention, and they could continue their normal daily activities. For ethical reasons, the program was also offered to the WLG following the post-intervention period of the SCG at T1.

### 2.10. Treatment Fidelity

To ensure consistency in administering the SCI, a written protocol for the intervention was developed. Additionally, all sessions were audio-recorded, and self-monitoring checking (Appendix A) was applied to assess the intervention sessions [69,70].

### 2.11. Data Collection

Self-administered questionnaires were assessed at three time points: T0, baseline; T1 (week 8), immediately after the intervention; and T2 (week 20), three months after the SCI ended. The first batch began in November 2021, followed by the second batch in April 2022 and the third batch in October 2022. The WLG was held immediately after the completion of the SCG. The first intervention in November 2021 had 10 participants and was initially conducted in a face-to-face mode until the sixth session. Owing to the worsening COVID-19 pandemic in January 2022, the intervention was switched to an online mode of delivery starting from the seventh session. From then on, all interventions were conducted online. The content of the online program was the same as the face-to-face group meetings.

### 2.12. Outcome Measures

#### 2.12.1. Primary Outcomes (Depression and Anxiety)

The 9-item Patient Health Questionnaire (PHQ-9) was used to assess the level of depression. Responses were provided on a scale of 0 to 3, with higher scores indicating a greater likelihood of being depressed [71]. The Cronbach’s alpha values were 0.795, 0.905, and 0.842 across the three time points, indicating good reliability. The 7-item General Anxiety Questionnaire (GAD-7) was used to measure the severity of anxiety [72]. Each item was scored from 0 to 3, with lower scores indicating a lower likelihood of being anxious. The GAD-7 showed excellent reliability across the three time points with Cronbach’s alpha values of 0.923, 0.920, and 0.889, respectively.

#### 2.12.2. Secondary Outcomes

The 16-item Daily Spirituality Experience Scale (DSES) was used to assess the ordinary experience of transcendence in daily life [73,74]. The 6-item State Hope Scale (SHS) was used to measure the participants’ subjective degree of hope [75,76]. The odd-numbered items measured pathway thinking (strategies designed to achieve goals) and even-numbered items measured agency thinking (determination directed toward goal attainment). The 10-item Meaning in Life Questionnaire (MLQ) was used to evaluate two constructs: the presence of meaning in life and the search for meaning in life [77,78]. The 10-item Rosenberg Self-esteem Scale was used to measure the participants’ self-esteem [79], and perceived social support was assessed with the 12-item Multidimensional Scale of Perceived Social Support (MSPSS), which included three domains: family, friends, and significant others [80]. The validated Chinese versions of these scales were used. Most of the measures demonstrated acceptable-to-excellent reliability across the three time points, except the SHS-Pathway subscale and the MLQ-Search subscale, which had relatively low reliability (for detailed scale characteristics and reliability coefficients, see the Appendix A).

### 2.13. Statistical Analysis

The data analyses were carried out using IBM SPSS Statistics 26, adopting a two-tailed significance level of *p* < 0.05. This study used an intention-to-treat (ITT) strategy, and any missing values in the outcome variables were replaced with the corresponding baseline values. The chi-square test and independent *t*-test were used to assess the baseline equivalence between SCG and WLG in the demographic characteristics and the outcome variables.

Repeated-measures ANOVAs and post hoc tests with Bonferroni corrections were used to examine within-group changes across the three time points, essentially, baseline vs. post-intervention and baseline vs. three-month follow-up, separately for SCG and WLG. This analysis also highlighted the temporal effects of the intervention in the SCG as opposed to any natural progression in the WLG.

To assess the program’s effectiveness, one-way ANCOVA was employed to examine the differences between the groups using change scores. Demographic factors such as age, gender, education, marital status, religious affiliation, employment status, past psychiatric treatment history, and the baseline values of each outcome variable (T0) were included as covariates to adjust for their potential influence on the outcomes. Change scores were calculated by subtracting the baseline values (T0) from the subsequent measurements (T1−T0). Interactions between demographic variables and group assignment were also added to the model to examine the moderation effects among the demographic variables.

## 3. Results

### 3.1. Demographic Characteristics

A total of 128 eligible adults were invited to this study (Figure 2). However, 71 individuals were excluded for various reasons, including not meeting the inclusion criteria or declining to participate. The remaining 57 participants completed the baseline assessment. Most of them were female (75.4%) aged 46–64 (71.9%). About half were married (47.4), held university-level education (63.2%), were employed full-time (43.9%), and had a history of psychiatric treatment (57.9%). The majority identified as Protestant Christian (86.0%), while a small portion identified as non-religious (10.5%).

The participants were randomized into the SCG (*n* = 28) and the WLG (*n* = 29). Some participants from both groups were absent or attended only a few sessions. The remaining 54 participants completed the assessments at all time points. Figure 2 shows the recruitment procedure and CONSORT flowchart. Baseline characteristics did not significantly differ between the two groups (Table 1).

The result of the baseline measurement indicated that there were no significant differences in the DSES, MLQ, RSES, MSPSS, PHQ-9, and GAD-7 scores. However, there was a significant difference in the SHS scores between the intervention group and the waitlist control group, with the intervention group having a higher mean score (SHS: *t* (55) = 2.083, *p* = 0.042, and mean difference = 4.683 (95% CI: 0.177–9.190); SHS-pathway: *t* (55) = 2.360, *p* = 0.022, and mean difference = 2.249 (0.339–4.159) (Table 2).

### 3.2. Diagnosis of Participants

In total, 20 out of 28 (71.44%) and 16 out of 29 (55.17%) participants in the intervention group and the WL control group did not have a formal psychiatric diagnosis (Appendix A); however, 15 out of 28 (62.1%) and 18 out of 29 (57.9%) in the intervention and WL control, respectively, had undergone psychiatric treatment before, including counseling. There was no statistically significant difference between the two groups in terms of the diagnosis of participants (χ^2^ (6) = 7.029; *p* = 0.318) and their history of psychiatric treatment (χ^2^ (1) = 0.422; *p* = 0.516).

### 3.3. Number of Sessions Attended

In total, 15 out of 28 (53.58%) and 17 out of 29 (58.62%) participants in the intervention group and the WL control group had full attendance; 17 out of 28 (60.71%) and 22 out of 29 (75.86%) participants in the intervention group and the WL control group had over 75% attendance (Appendix A). There was no statistically significant difference in the distribution of session attendance between the two groups (χ^2^ (7) = 8.086; *p* = 0.325). 

### 3.4. Changes in Outcomes for Within-Group Comparisons

The intervention had significant immediate and sustained effects on multiple psychological outcomes. Specifically, in the SCG, there was a large and significant reduction in depression immediately post-intervention (Cohen’s *d* = −1.452; *p* < 0.001) that persisted at the three-month follow-up (Cohen’s *d* = −1.325; *p* < 0.001). Anxiety levels also significantly decreased, with a large effect size post-program (Cohen’s *d* = −1.124; *p* < 0.001), which was maintained at follow-up (Cohen’s *d* = −1.045; *p* < 0.001). Notably, there was no effect from T0 to T1 for the WLG for either depressive symptoms or anxiety levels. Increases in spiritual experience were notable immediately after the intervention with a small effect size (Cohen’s *d* = 0.308; *p* < 0.01), although this was not sustained at follow-up. Participants reported more hope at both the immediate (Cohen’s *d* = 0.633; *p* < 0.01) and follow-up assessments (Cohen’s *d* = 0.588; *p* < 0.05). Interestingly, while the SCG did not show significant changes in meaning in life overall, the WLG displayed a significant increase in this measure post-program (Cohen’s *d* = 0.597; *p* < 0.01). Self-esteem in the SCG was significantly higher immediately after the program and at follow-up (Cohen’s *d* = 0.503 and 0.627, respectively, both *p* < 0.01). Moreover, perceived social support was significantly stronger post-intervention in the SCG with moderate effect sizes (Cohen’s *d* = 0.515; *p* < 0.05), specifically that from friends (Cohen’s *d* = 0.540; *p* < 0.01), but these effects did not persist at follow-up (for actual values and statistics, see the Appendix A).

### 3.5. Changes in Outcomes for Between-Group Comparisons

Table 3 shows that the SCG reported significantly lower scores in depression than the WLG after participation in the SCI (Cohen’s *d* = −1.801; *p* < 0.001). These change scores reflect a significant reduction in depressive symptoms from the baseline assessment (T0) to the post-intervention evaluation (T1).

Similarly, anxiety scores were significantly lower in the SCG compared with the WLG after participation in the SCI (Cohen’s *d* = −1.605; *p* < 0.001). These change scores demonstrate a substantial decrease in anxiety levels from the baseline assessment (T0) to the post-intervention evaluation (T1).

Furthermore, the SCG showed a significant improvement in spiritual experience compared with the WLG at T1 (Cohen’s *d* = 0.879; *p* < 0.01). These change scores reflect an enhancement in spiritual experience from the baseline (T0) to the post-intervention assessment (T1).

The SCG also reported a significantly higher degree of hope after completing the SCI than the WLG (Cohen’s *d* = 1.298; *p* < 0.001). These change scores represent a marked increase in hope from the baseline (T0) to the post-intervention (T1) evaluation.

Additionally, the SCG showed significantly higher scores in agency thinking and pathway thinking than the WLG at T1 (Cohen’s *d* = 1.424; *p* < 0.001) and (Cohen’s *d* = 0.960; *p* < 0.01), respectively. These change scores reflect the considerable improvement in hope from the baseline assessment (T0) to the post-intervention evaluation (T1).

The SCG also reported higher scores in the presence subscale of the MLQ than the WLG at T1 (Cohen’s *d* = 0.585; *p* < 0.05). These change scores represent the substantial rise in the meaning in life presence subscale from the baseline assessment (T0) to the post-intervention evaluation (T1).

Finally, the SCG reported higher self-esteem than the WLG at T1 after undergoing the SCI (Cohen’s *d* = 0.654; *p* < 0.05). These change scores represent the notable growth in self-esteem from the baseline assessment (T0) to the post-intervention evaluation (T1).

Interestingly, the between-group difference in MSPSS scores was not significant at T1, including the sub-domain scores for family, friends, and significant others.

### 3.6. Moderation Effect

Table 4 indicates significant effects between age and group interaction on several outcome variables. For the primary outcomes, there was a significant age-by-group interaction in depression scores (PHQ-9), such that the intervention had a greater impact in reducing depressive symptoms among the older participants compared with the younger participants (mean difference = −4.746; *p* = 0.004; Cohen’s *d* = −1.221). However, the interaction effect was not significant for anxiety scores (GAD-7).

For the secondary outcomes, significant age-by-group interactions were observed for several measures. The intervention had a stronger positive effect among older participants compared with younger participants on hope scores (SHS: mean difference = 8.669; *p* = 0.036; Cohen’s *d* = 1.231), agency thinking (SHS-Agency: mean difference = 4.912; *p* = 0.013; Cohen’s *d* = 1.195), perceived social support (MSPSS: mean difference = 11.135; *p* = 0.033; Cohen’s *d* = 1.053), and perceived support from friends (MSPSS-Friend: mean difference = 4.085; *p* = 0.046; Cohen’s *d* = 0.904). However, there were no significant age-by-group interactions found for spiritual experiences (DSES), meaning in life (MLQ), self-esteem (RSES), or perceived support from family and significant others (for the marginal means of the two groups, see the Appendix A) (see Appendix A for code used for experimental implementation).

### 3.7. Record of Daily Activities in the Mobile App

Only a few participants engaged in the mobile apps and most of them did not use it; thus, we were not able to assess the participants’ daily spiritual activities through the apps.

### 3.8. Sensitivity Analysis

A sensitivity analysis was conducted to compare the results of the ITT analysis and per-protocol (PP) analysis. The ITT analysis included all participants (*n* = 57), while the PP analysis included those who completed the program (*n* = 26 in the SI and *n* = 28 in the WL). The statistical significance of the outcome measurements did not differ between the two analysis methods (Appendix A).

### 3.9. Clinical Significance

In total, 24 out of 28 (85.71%) participants from the SCG had at least a 5-point reduction in their PHQ-9 scores from baseline (T0) to post-intervention (T1). By contrast, only 1 out of the 29 participants (3.45%) in the WLG experienced a 5-point reduction in their PHQ-9 scores during the same period (Appendix A). Furthermore, 20 out of 29 (68.97%) participants from the WLG also attained at least a 5-point reduction in their PHQ-9 scores immediately after the intervention was completed.

## 4. Discussion

The present study examined the effect of an SCI on various psychological outcomes in adults with mild or moderate depressive symptoms. The results of the within-group comparisons and the findings in Table 3 indicate that the SCG experienced significant improvements in depressive symptoms, anxiety levels, spiritual experience, hope, self-esteem, and perceived social support compared with the WLG. The positive effects could still be observed in the follow-up measurement (T2), especially for depressive symptoms and anxiety levels. Other improvements were also maintained at follow-up within the SCG, such as spiritual experience, hope, agency thinking, pathway thinking, and self-esteem.

The participants in both the SCG and the WLG reported a significant reduction in depressive symptoms and anxiety after participating in the SCI. This aligns with prior research showing that spiritual interventions can decrease depression and anxiety [81,82,83]. Potential mechanisms include peer support and the transcendence of suffering, which can contribute to meaning and connectedness to God [41,84]. This discovered meaning offers comfort, hope, enhanced coping, and reduced distress [14]. Interestingly, the participants’ increased sense of meaning and closeness to God was not accompanied by a corresponding increase in their perceived social support, as might be expected. This suggests that the mechanism linking the intervention to the outcomes of meaning and connectedness to God was more closely tied to spiritual and transcendent experiences rather than social support mechanisms. Spiritual practices like meditation, prayer, and hymn singing may also promote relaxation, emotional regulation, and cognitive reframing, leading to symptom alleviation [85]. Importantly, these positive changes were sustained at the 3-month follow-up, with large effect sizes indicating substantial and lasting improvements in depression and anxiety. These findings highlight the potential of these mechanisms within the SCI to significantly impact participants’ mental health.

Participants in both the SCG and the WLG reported a significant increase in spiritual experience after participating in the spiritual connectivity program. Previous studies of spiritual intervention on clients with cancer also showed spiritual growth and improved spiritual well-being after the intervention [86,87,88]. The observed relationships suggest that an increase in spiritual engagement could potentially benefit mental health by notably decreasing symptoms of depression and anxiety. Looking ahead, future research endeavors could delve into the enduring effects of the SCI on mental health outcomes over an extended period. Longitudinal studies have the potential to elucidate the sustained impact of the intervention on aspects like depression and anxiety, offering insights into the long-term efficacy of SCI in fostering improved mental well-being.

The participants in both the intervention (SCG) and waitlist control (WLG) groups reported a significant increase in hope after the SCI, which was sustained at the 3-month follow-up. The moderate effect sizes indicate a meaningful improvement in hope. Prior research has shown that spiritual interventions can promote hope in inpatient settings [86,89]. Spirituality fosters hope through trust in God and the incorporation of faith [90], with the Christian doctrine emphasizing eternal life. Additionally, both groups exhibited increases in agency thinking and pathway thinking, which are components of hope. These findings suggest that the SCI had a positive and lasting impact on participants’ sense of hope.

While the intervention (SCG) group did not report significant changes in meaning in life, the waitlist control (WLG) group showed a significant increase, particularly on the presence subscale, with a moderate effect size. This suggests that the SCI had a positive impact on the sense of meaning in life for the WLG participants. Changes in meaning in life involve complex processes like exploring sources of meaning, connecting the past and present, and reflection [88,91]. To enhance the intervention’s impact, strategies could include increasing intensity through additional sessions or an extended program, providing attendance incentives to boost engagement, and implementing reminders for consistent practice outside of sessions. These approaches may allow participants to further explore and understand meaning in life, leading to more prominent and sustained effects.

The participants in both the SCG and the WLG reported significantly higher self-esteem after completing the SCI, with sustained improvements at the 3-month follow-up. The moderate effect sizes indicate a meaningful enhancement in self-esteem, consistent with prior research [92,93,94]. Spirituality has been linked to higher self-esteem [95], and the spiritual practice of regular prayer is associated with increased self-esteem [94]. Prayer’s interaction with a divine/higher power, combined with a belief in divine support, can impact feelings of self-worth [96]. Furthermore, an SCI’s enhancement of spiritual growth and positive relationships with God can provide emotional support, further boosting self-esteem [97].

An analysis of the interactions between demographic variables and the SCI revealed that age moderated several outcomes. The impact on depression, hope, agency thinking, perceived social support, and support from friends varied across age groups. Notably, older adults appeared to benefit more from the intervention, echoing previous findings on age as a moderator of spiritual programs [98,99]. This prompts us to consider modifying the program to ensure that younger adults can also experience greater benefits. Future research could explore the specific factors contributing to the differential effects across age groups, such as addressing participants’ intervention expectations during recruitment. This knowledge would help tailor the spirituality program to better accommodate the needs and preferences of younger adults, enabling them to experience significant benefits as well.

Regarding the sensitivity analysis, the results indicated that the statistical significance of the outcome measures did not differ between the intention-to-treat (ITT) analysis and the per-protocol (PP) analysis. This suggests that the findings are robust and not heavily influenced by dropouts or non-compliance with the program. Both analysis methods yielded consistent results, supporting the validity of this study’s findings.

In terms of clinical significance, a high proportion of participants in the SCG (85.71%) achieved clinically significant reductions in depressive symptoms (≥5-point decreases in the PHQ-9) from baseline to post-intervention. By contrast, only 3.45% of the waitlist control group (WLG) reached this level of improvement during the same period. Additionally, a substantial 68.97% of WLG participants also exhibited reduced depressive symptoms after the intervention. These findings highlight the clinical significance of the SCI in alleviating depressive symptoms, especially in the intervention group. A 5-point or greater reduction in PHQ-9 is considered clinically meaningful [100,101,102].

In our study, we also recognized the potential negative impacts of spirituality and religiosity and the uniqueness of each individual. To safeguard participants, our carefully selected facilitators underwent comprehensive training and adhered strictly to ethical guidelines. They were experienced in mental health and group dynamics, ensuring a safe environment. Through ongoing training, they honed their skills while remaining sensitive to participants’ diverse beliefs. Upholding ethical standards and clear communication, we aimed to prevent any adverse effects, fostering trust, respect, and well-being throughout the intervention.

The strength of this study lies in its demonstration of the broad efficacy in enhancing diverse psychological outcomes, indicating its comprehensive approach to mental health. The findings suggest that the intervention may work by cultivating a sense of meaning and connection to a higher power/God, rather than through social support mechanisms. The identification of age as a moderating factor provides valuable insights for tailoring the intervention to different age groups effectively. A significant proportion of SCI participants achieved substantial reductions in depressive symptoms, highlighting the interventions’ practical relevance.

### 4.1. Limitations

Several limitations are noted. First, the researchers’ motivations and involvement may have influenced the program’s effectiveness, so future research could explore implementation by different individuals/organizations to validate generalizability. Second, the small sample size may also limit statistical power and generalizability. Third, the longer-term effects of the intervention need to be examined to provide stronger evidence of efficacy. Fourth, the use of subjective self-report measures may have been influenced by social desirability. Fifth, the reliability of the SHS-Pathway and MLQ Search subdomain was relatively low; these issues were previously discussed by the authors [76,78]. Nevertheless, it is important to note that both scales are commonly used. Sixth, the waitlist control design may have introduced biases or confounding factors. Seventh, the heterogeneity in the mode of delivery of the intervention may have affected the consistency of the mode of intervention. Finally, the characteristics of the participants need to be considered. More than half of the participants were female and over 80% of the participants affiliated themselves with Protestant Christianity. Their perception and response to faith-based spirituality may be different from individuals with non-Christian backgrounds. Thus, the results need to be interpreted with caution.

### 4.2. Implications

The positive impact of the SCI suggests that integrating spirituality into therapeutic interventions may be beneficial. Mental health professionals could consider incorporating spiritual elements to enhance outcomes. SCI shows potential for application across diverse mental health issues, with components like forgiveness and freedom holding relevance for trauma and post-traumatic stress disorders, while a focus on hope and gratitude could be beneficial for individuals with anxiety disorder. SCI could potentially be adapted to different religious or spiritual traditions with universal themes of forgiveness, suffering, hope, and gratitude. Healthcare professionals need to be flexible and respect clients’ cultural needs when implementing the intervention.

In future studies, exploring the potential reach of the SCI to non-local and younger populations within the church community could offer valuable insights into the intervention’s applicability across diverse demographics. Given the inclusive nature of church settings, where activities cater to various linguistic and age groups, investigating the adaptability of the intervention to these subpopulations could enrich the understanding of its broader effectiveness and relevance in mental healthcare.

Larger-scale studies with diverse populations and longer follow-ups would improve generalizability and evaluate long-term effectiveness. Future research could use objective measures or additional data sources to strengthen validity. Qualitative explorations of participants’ perspectives would provide a more comprehensive understanding of the underlying mechanisms and how individual characteristics interact with the program and influence outcomes. Further research could investigate the benefits of spirituality programs in other mental health conditions and explore optimal dosage and duration.

## 5. Conclusions

In conclusion, these findings suggest that our SCI had positive effects on hope, self-esteem, depression, and anxiety among participants. While the improvements in spiritual experience, meaning in life, and perceived social support were limited in duration, the changes in hope, self-esteem, depressive symptoms, and anxiety level were sustained over the three-month follow-up period. These results highlight the potential benefits of incorporating spirituality and connectivity into therapeutic interventions. The program is non-invasive, relatively low-cost, and easily accessible, and it may be perceived as less stigmatized than conventional mental health treatments. With a standardized protocol in place, there is potential for the program to be implemented successfully in the community by pastors, clergy members in the church, and program workers in faith-based non-government organizations. This could contribute to alleviating the strain on the public healthcare system. Future research and collaboration with these stakeholders would be valuable in establishing the feasibility and impact of such community-based implementation.

## Figures and Tables

**Figure 1 healthcare-12-01604-f001:**
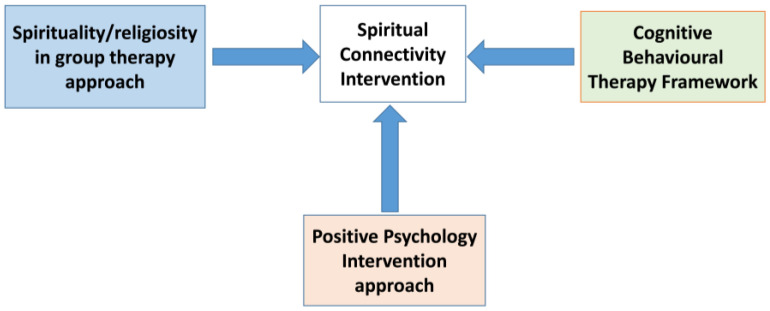
Spiritual connectivity intervention framework [35].

**Figure 2 healthcare-12-01604-f002:**
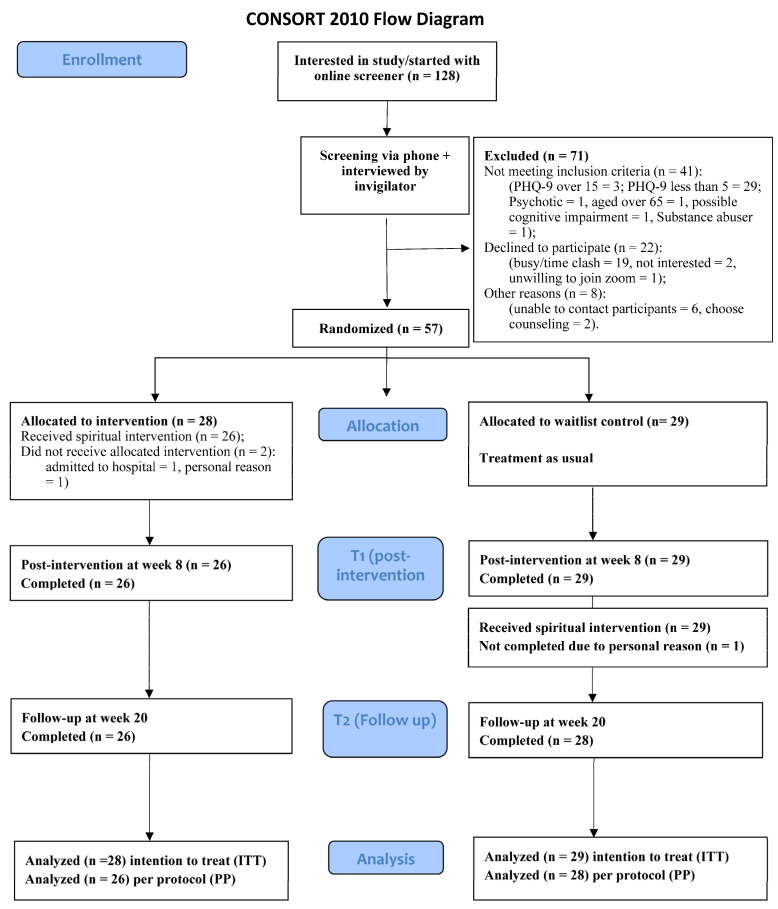
Flow chart of participant recruitment, distribution, and follow-up.

**Table 1 healthcare-12-01604-t001:** Demographic characteristics.

Variables	Total (*N* = 57)	Intervention Group	WL Control Group	*p*-Value *
	(*N*, %)	(*n* = 28)	(*n* = 29)	
Gender				0.49
Male	14 (24.6)	8 (28.6)	6 (20.7)	
Female	43 (75.4)	20 (71.4)	23 (79.3)	
Age group (years)				0.162
18–25	3 (5.3)	1 (3.6)	2 (6.9)	
26–35	4 (7.0)	4 (14.3)	0	
36–45	9 (15.8)	6 (21.4)	3 (10.3)	
46–55	21 (36.8)	9 (32.1)	12 (41.4)	
56–64	20 (35.1)	8 (28.6)	12 (41.4)	
Religion				0.367
Protestant Christian	49 (86.0)	25 (89.3)	24 (82.8)	
Roman Catholic	2 (3.5)	0	2 (6.9)	
Non-religious	6 (10.5)	3 (10.7)	3 (10.3)	
Educational level				0.227
Primary	2 (3.5)	0	2 (6.9)	
Secondary	19 (33.3)	12 (42.9)	7 (24.1)	
Undergraduate	27 (47.4)	13 (46.4)	14 (48.3)	
Postgraduate	9 (15.8)	3 (10.7)	6 (20.7)	
Marital status				0.17
Single	22 (38.6)	11 (39.3)	11 (37.9)	
Married	27 (47.4)	14 (50.0)	13 (44.8)	
Divorced	4 (7.0)	3 (10.7)	1 (3.4)	
Widowed	4 (7.0)	0	4 (13.8)	
Occupation				0.504
Full-time student	5 (8.8)	2 (7.1)	3 (10.3)	
Full-time employee	25 (43.9)	14 (50.0)	11 (37.9)	
Part-time employee	8 (14.0)	4 (14.3)	4 (13.8)	
Unemployed	6 (10.5)	3 (10.7)	3 (10.3)	
Retired	9 (15.8)	2 (7.1)	7 (24.1)	
Others	4 (7.0)	3 (10.7)	1 (3.4)	
Treatment				0.516
Yes	33 (57.9)	15 (62.1)	18 (57.9)	
No	24 (42.1)	13 (37.9)	11 (42.1)	

* A *p*-value of <0.05 was considered statistically significant.

**Table 2 healthcare-12-01604-t002:** The outcome measures at baseline (T0).

Variables	Total (*N* = 57) (Mean, SD)	Intervention Group (*n* = 28) (Mean, SD)	WL Control Group (*n* = 29) (Mean, SD)	*t*-Value	*p*-Value *	Mean Difference (95% CI)
DSES	56.51 (12.74)	57.39 (12.44)	55.66 (13.19)	0.511	0.611	1.738 (−5.072–8.547)
SHS	25.44 (8.74)	27.82 (8.24)	23.14 (8.72)	2.083	0.042 *	4.683 (0.177–9.190)
SHS-Agency	11.37 (5.43)	12.61 (5.05)	10.17 (5.6)	1.721	0.091	2.435 (−0.401–5.271)
SHS-Pathway	14.07 (3.74)	15.21 (3.55)	12.97 (3.64)	2.360	0.022 *	2.249 (0.339–4.159)
MLQ	44.21 (10.25)	46.89 (10.24)	41.62 (9.73)	1.993	0.051	5.272 (−0.030–10.575)
MLQ-Presence	19.44 (6.78)	20.57 (6.64)	18.35 (6.85)	1.246	0.218	2.227 (−1.356–5.809)
MLQ-Search	24.77 (6.44)	26.32 (4.64)	23.28 (7.59)	1.821	0.073	3.046 (−0.293–6.384)
RSES	24.88 (5.04)	25.21 (5.06)	24.55 (5.09)	0.493	0.624	0.663 (−2.031–3.357)
MSPSS	50.32 (13.97)	51.82 (14.36)	48.86 (13.67)	0.797	0.429	2.959 (−4.483–10.402)
MSPSS-Family	15.40 (5.45)	16.64 (5.36)	14.21 (5.36)	1.716	0.092	2.436 (−0.409–5.281)
MSPSS-Friend	17.09 (5.16)	17.14 (5.67)	17.03 (4.71)	0.079	0.938	0.108 (−2.654–2.871)
MSPSS-Sig. others	17.83 (5.36)	18.04 (5.22)	17.62 (5.57)	0.290	0.773	0.415 (−2.452–3.282)
PHQ-9	9.51 (4.26)	9.07 (4.62)	9.93 (3.92)	−0.759	0.451	−0.860 (−3.130–1.411)
GAD-7	8.56 (4.70)	8.21 (4.90)	8.90 (4.56)	−0.544	0.588	−0.682 (−3.195–1.830)

* A *p*-value of <0.05 was considered statistically significant. The *p*-values were computed using independent *t*-tests.

**Table 3 healthcare-12-01604-t003:** The between-group differences in changes in depression, anxiety, spiritual experience, hope, meaning in life, self-esteem, and perceived social support at T1 in the intervention and waitlist control groups: intention-to-treat analysis.

	Intervention vs. Waitlist Control	
Measures	Mean Difference [95% CI]	Cohen’s *d*
**Primary outcomes**		
Depression (PHQ-9)	−6.184 [−8.046, −4.322]	−1.801 ***
Anxiety (GAD-7)	−4.858 [−6.500, −3.216]	−1.605 ***
**Secondary outcomes**		
Spiritual experience (DSES)	6.403 [2.454, 10.351]	0.879 **
Hope (SHS)	8.720 [5.077, 12.363]	1.298 ***
Agency thinking (SHS-Agency)	5.341 [3.307, 7.375]	1.424 ***
Pathway thinking (SHS-Pathway)	3.386 [1.474, 5.298]	0.960 **
Meaning in life (MLQ)	6.394 [0.234, 12.555]	0.563 *
Presence of meaning in life (MLQ-Presence)	3.863 [0.285, 7.442]	0.585 *
Search of meaning in life (MLQ-Search)	1.877 [−1.008, 4.761]	0.353
Self-esteem (RSES)	1.777 [0.304, 3.249]	0.654 *
Perceived social support (MSPSS)	5.646 [−0.385, 11.678]	0.508
Perceived support from family (MSPSS-Family)	2.223 [−0.337, 4.783]	0.471
Perceived support from friends (MSPSS-Friend)	1.741 [−0.703, 4.186]	0.386
Perceived support from significant others (MSPSS-Sig.)	1.555 [−0.447, 3.556]	0.420

Note. One-way ANCOVA was used to examine the between-group differences. Mean differences and effect size (Cohen’s *d*) were computed from estimated marginal means. * *p* < 0.05, ** *p* < 0.01, and *** *p* < 0. 001.

**Table 4 healthcare-12-01604-t004:** Mean differences in the outcome variables by intervention group (SCG) age subgroups post-intervention (T1).

Outcome Variables	Mean Difference (SE)	*p*-Value (Age * Group)	Cohen’s *d* (95% CI)
**Primary outcomes**			
Depression (PHQ-9)	−4.746 (1.612) *	0.004	−1.221 (−0.349, −2.093)
Anxiety (GAD-7)	−4.447 (1.647)	0.077	−1.120 (−0.254, −1.986)
**Secondary outcomes**			
Spiritual experience (DSES)	8.213 (4.231)	0.457	0.805 (−0.046, 1.656)
Hope (SHS)	8.669 (2.920) *	0.036	1.231 (0.359, 2.104)
Agency thinking (SHS-Agency)	4.912 (1.705) *	0.013	1.195 (0.324, 2.065)
Pathway thinking (SHS-Pathway)	3.757 (1.463)	0.193	1.065 (0.202, 1.928)
Meaning in life (MLQ)	3.680 (4.642)	0.293	0.329 (−0.508, 1.166)
Presence of meaning in life (MLQ-Presence)	4.195 (3.081)	0.164	0.565 (−0.278, 1.407)
Search for meaning in life (MLQ-Search)	−0.515 (2.438)	0.816	−0.088 (−0.922, 0.747)
Self-esteem (RSES)	2.163 (1.557)	0.086	0.576 (−0.267, 1.419)
Perceived social support (MSPSS)	11.135 (4.386) *	0.033	1.053 (0.191, 1.915)
Perceived support from family (MSPSS-Family)	3.833 (1.832)	0.060	0.868 (0.014, 1.721)
Perceived support from friends (MSPSS-Friend)	4.085 (1.873) *	0.046	0.904 (0.049, 1.760)
Perceived support from significant others (MSPSS-Sig.)	3.217 (1.529)	0.142	0.873 (0.019, 1.726)

* *p* < 0.05. Younger = 18–45 years (*n* = 11); older = 46–64 years (*n* = 17).

## Data Availability

Data are contained within the article or Appendix A. The participants of this study did not give written consent for their data to be shared publicly, so, due to the sensitive nature of the research, supporting data are not available.

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
