# Peer review of "Spiritual Connectivity Intervention for Individuals with Depressive Symptoms: A Randomized Control Trial"

_healthcare, 2024, doi:10.3390/healthcare12161604_

Round 1

Reviewer 1 Report

Comments and Suggestions for Authors

Dear Authors:

The matter of your topic is interesting; although there is some novelty in this study, taking a look at the whole topic, there is a lot of literature in this field, and your result shows the same as others; as a general question, what does your work contribute to current literature?

Given the specificity of the topic and terms of religion, I have noticed that you asked participants about their religion. Did you check if participants are also part of a church or any related to spiritual groups? This is a matter of the fact that if they are, either they resonated better because of the new group they were involved with or other factors.

Statistically, the data looks nice, although you haven’t satisfied the number of participants.

The matter of the topic, specifically in the context of "Christianity," influences and restricts the intervention. Although this is a controversial intervention and does not encompass standard evidence on the topic, I would like to ask the authors how this work may contribute to scientific literature, specifically for non-local and new generations.

The method proposed by the authors includes potential detrimental effects such as negative religious coping, treatment refusal, intolerance, and negative beliefs, which can harm mental health. How do the authors intend to address this matter in their work? (For reference, see A. Moreira-Almeida et al., Samuel R. Weber et al., R. Page et al., G. Lucchetti et al.)

The authors mentioned, "Clergy members have long provided counselling and care to church members...," which, although cited in other literature (see reference [15]), does not resonate with the academic and formal background necessary for this purpose. Specifically, we already know that in such studies, there is a lack of formal education on how to address these aspects in clinical practice (see Naziha S. ; Aleati et al., A. Moreira-Almeida et al.).

The taxonomy of "Forgiveness" [Line 73] in the context of depressive symptoms is highly case-dependent. This needs to be defined on a case-by-case basis and personalized rather than generalized.

Did the authors have any criteria for the "social identity" of the participants as a key element of such therapies? If so, how and which method was implemented?

The authors mentioned the required number (through power analysis) to obtain the necessary power. However, it appears to me that the number of participants is insufficient. If I am mistaken, please let me know; otherwise, how do the authors justify the lack of sufficient power?

With 8 out of 57 participants declaring different religions for such treatment, do the authors not think this may impact the results?

Given the variety of activities in your treatment, have you tried to check which factor in terms of the participants had more impact on them or not?

Taking into account your Table S2, why did you not give your participants more options?

You have some participants who did not attend all sessions (right?). Did you use their data also in the analysis? Or not?

Did you consider the factor of age? Are there any differences by age?

Comments on the Quality of English Language

Minor edits are needed, and transitions between some sentences are not clear. overall English level is good, but some minor needs are there

Author Response

We would like to express our sincere gratitude to reviewer for your thorough reading of this manuscript and for the thoughtful and constructive comments, which helped improve the quality of the paper. We appreciate the time and effort that you dedicated to providing feedback on it. After carefully addressing all the comments, the refinements are made in the revised manuscript, which is highlighted in red.  The detailed responses are summarized in Point-by-point response to Comments and Suggestions for Authors.

Point-by-point response to Comments and Suggestions for Authors

Comments 1:  The matter of your topic is interesting; although there is some novelty in this study, taking a look at the whole topic, there is a lot of literature in this field, and your result shows the same as others; as a general question, what does your work contribute to current literature?

Response 1: Thank you for your feedback and comments on our study. We appreciate your recognition of the topic's interest and the novelty in our study. We understand that there is existing literature in this field, and it is essential to contextualize our work within the broader research landscape.

Our study contributes to the current literature and clinical practice. Firstly, it explores the effectiveness of a community based spiritual connectivity intervention program specifically tailored for individuals with depressive symptoms. While there are various interventions available for depression, our program uniquely focuses on Christian faith and connectedness as key components. This specific emphasis on connectedness sets our study apart from other interventions in the field.

Secondly, the study addresses the challenge of low awareness and stigma associated with mental illness, which often hinders individuals from seeking professional help (Li et al., 2018). By implementing a community-oriented intervention, we aim to bridge the service gap and provide support to a difficult-to-reach population. This aspect of our study is particularly significant as it targets early identification and intervention for individuals who may not have access to or engage with traditional mental health services. Thus, it can significantly ease the burden on public healthcare system. Revision in the introduction is made in P.1 and 2 Line 33 and 45 respectively under Section 1 Introduction of the manuscript:

“Depression is one of the most under-reported and under-treated mental health issue (Kessler & Bromet, 2013). This may be due to ignorant or lack of knowledge about their health problem. People’s belief in mental health or stigmatization might also hinder their help-seeking behaviour (Li et al., 2018).

                What distinguishes our study is the explicit focus on fostering connectedness, a vital component that sets it apart from other interventions in this domain. By implementing a community-oriented approach, we aim to bridge the existing service gap and extend support to a segment of the population that is often hard to reach. This initiative is particularly significant as it aims to facilitate early identification and intervention for individuals who might otherwise lack access to or engagement with conventional mental health services, potentially alleviating the strain on the public healthcare system.”

Furthermore, it is mentioned in P.3 line 125 under Session1.2 Therapeutic components “there is a dearth of studies using rigorous methodological designs, such as randomized controlled trials (RCTs), to evaluate the efficacy of spiritual interventions for individuals with depression.” By incorporating randomized controlled trial (RCT) into the study design, this research aims to contribute uniquely to the field by rigorously evaluating the effectiveness of the spiritual connectivity intervention in a controlled and scientifically robust manner. We believe that our study adds value and unique insight to the current literature in this field.

Comments 2:  Given the specificity of the topic and terms of religion, I have noticed that you asked participants about their religion. Did you check if participants are also part of a church or any related to spiritual groups? This is a matter of the fact that if they are, either they resonated better because of the new group they were involved with or other factors.

Response 2: We appreciate the reviewer's comment and acknowledge the importance of considering participants' religious affiliation and involvement in spiritual groups in studies related to spirituality and mental health interventions. While we did collect information on participants' religious affiliation, we did not specifically inquire about their involvement in churches or spiritual groups.

We recognize that participation in such groups can influence individuals' experiences and perceptions of spirituality, potentially affecting their response to the intervention. Participants' affiliation with religious or spiritual groups could impact how they engage with a new group or setting, potentially leading to a stronger connection or resonance because of shared beliefs or experiences.

It has been acknowledged in the last session in limitation in our study “Finally, the predominance of female and Protestant Christian participants suggests caution in generalizing the results to other populations. Thus, the results need to be interpreted with caution.”  We have revised the wordings in P. 15 line 575 under Section 4.1 Limitations of the manuscript as below to make it more explicit:

“Finally, characteristics of the participants need to be considered. More than half of the participants were females and over 80% participants affiliated themselves as Protestant Christianity. Their perception and response to faith-based spirituality may be different from individuals with non-Christian background. Thus, the results need to be interpreted with caution”

Moving forward, we will consider the specific aspects of participants' church involvement, such as participation in prayer groups, Bible study groups, or service activities, to gain a more comprehensive understanding of how these factors may have influenced their experiences within the intervention. By acknowledging and addressing these considerations, we aim to enhance the depth and relevance of our study findings.

Comments 3:  Statistically, the data looks nice, although you haven’t satisfied the number of participants.

Response 3: Thank you for your review and valuable feedback. We appreciate your observation regarding the sample size. While we aimed for a sample size of 58, we ended up with 57 participants in the study. We acknowledge the importance of meeting the intended sample size for statistical robustness. For a detailed response addressing this concern, we kindly ask the reviewer to refer to our response provided in Comment 9. Should you require any further clarification or additional information, please do not hesitate to reach out. Your insights are greatly appreciated in our ongoing efforts to enhance the quality and robustness of our research.

Comments 4: The matter of the topic, specifically in the context of "Christianity," influences and restricts the intervention. Although this is a controversial intervention and does not encompass standard evidence on the topic, I would like to ask the authors how this work may contribute to scientific literature, specifically for non-local and new generations.

Response 4: In response to the reviewer's comments, we appreciate the insightful feedback provided. The influences of the topic, particularly centered around "Christianity," on the intervention studied is indeed a crucial aspect to consider. We acknowledge the controversial nature of the intervention and would like to emphasize there is a growing interest and handful of research studies on spiritual intervention.

Regarding how our work may contribute to scientific literature, especially for broader audiences and future generations, we believe our research offers a unique perspective on the intersection of Christian faith, connectivity and mental health care. By delving into this controversial yet significant area, we aim to shed light on potential avenues for integrating faith-based approaches into therapeutic interventions.

Our study builds upon the existing evidence-based literature on spiritual interventions, such as the works by Anderson et al. (2015), Gonçalves et al. (2015), Captari et al. (2018), and Bouwhuis-Van Keulen et al. (2023). These studies have provided valuable insights into the effectiveness of faith-adapted psychological therapies and religious/spiritual interventions in mental health care.

Through our research, we strive to contribute to a deeper understanding of how religious and spirituality-based therapies can complement standard treatments in mental health care settings. By conducting a comprehensive analysis of the impact of these interventions, we hope to offer new perspectives that can benefit both current and future generations of practitioners and researchers in the field.

For reviewer’s concern on non-local and new generations, our study primarily involved participants who could speak Cantonese and did not encompass non-local individuals, we recognize the potential for the intervention to reach a diverse range of populations, including non-local and younger groups, within church settings. In future studies, we aim to explore the adaptability and generalizability of the SCI to a broader demographic, including non-local and younger populations. By investigating the intervention's reach across different subpopulations within the church community, we aspire to enhance the applicability and effectiveness of faith-based approaches in mental health care. The following paragraph is added to P.15 line 589 under Section 4.2 Implications of the manuscript:

“In future studies, exploring the potential reach of the SCI to non-local and younger populations within the church community could offer valuable insights into the intervention's applicability across diverse demographics. Given the inclusive nature of church settings, where activities cater to various linguistic and age groups, investigating the adaptability of the intervention to these subpopulations could enrich the understanding of its broader effectiveness and relevance in mental health care.”

Thank you for highlighting these important considerations, and we are committed to addressing them in our work to enhance its relevance and contribution to the scientific literature.

Comments 5:  The method proposed by the authors includes potential detrimental effects such as negative religious coping, treatment refusal, intolerance, and negative beliefs, which can harm mental health. How do the authors intend to address this matter in their work? (For reference, see A. Moreira-Almeida et al., Samuel R. Weber et al., R. Page et al., G. Lucchetti et al.)

Response 5: We appreciate the thoughtful insights provided by the reviewer regarding the potential risks associated with spiritual interventions, including negative religious coping, treatment refusal, intolerance, and the development of negative beliefs. These concerns are pivotal in ensuring the ethical and responsible conduct of our work.

We recognize there may be negative effect of religiosity/spirituality (R/S). There are positive and negative effect of R/S. Most of the research studies have consistently demonstrated R/S positive impact on health, but may also less frequently have a negative impact (Moreira-Almeida & Bhugra, 2021, p. 21). Understand the positive and negative effect of R/S can help to provide evidence-based practice and improve patient outcomes. Healthcare professionals need to be sensitive to clients’ spiritual needs as they can influence the progress of their illness.  We address the issue by adding the following in P.2, line 58 under Section 1.1 Spirituality, religion, and depression of the manuscript:

Over the past decades, mental health professionals have displayed an increasing interest in incorporating psychosocial and spiritual care for mental health recovery (Fallot, 2001; Gomi et al., 2014). While studies generally suggest positive associations between religion/spirituality and depression, negative effects can also arise. Individuals may experience guilty and shame when they fail to meet their religious/spiritual community’s behavioral expectations, which can lead to anxiety and depression (Kao et al., 2020).  Engaging in negative religious coping mechanisms, such as blaming God for personal hardships, has been linked to increased depressive symptoms (Smith et al., 2003). Healthcare professionals need to be sensitive to clients’ spiritual needs as they can influence the progress of their illness.

Please be assured that the spiritual intervention was carefully designed to promote positive coping, tolerance, and positive beliefs. All potential participants were screened and assessed to identify at their risk of negative outcomes. We also emphasized the importance of respecting individual differences in religious and spiritual beliefs, and ensure that our interventions were adaptable to suit different belief systems. We prioritize transparent communication with participants about the complementary nature of spiritual interventions alongside conventional treatment. The spiritual connectivity intervention in this study was conducted by three facilitators. All the intervention groups was conducted by the one of the authors who is an experienced psychiatric nurse. The waitlist control groups were conducted by an experienced social worker and an experienced social worker. All the facilitators got extensive experience in serving clients with mental health issues and leading therapeutic groups. There were regular meetings to enhance our communication on the treatment delivery.

The following paragraph is added to P.14 line 547 under Section 4. Discussion of the manuscript to highlight the transparency & stringency of the study:

In our study, we also recognized the potential negative impacts of spirituality and religiosity, and the uniqueness of each individual. To safeguard participants, our carefully selected facilitators underwent comprehensive training and adhered strictly to ethical guidelines. They were experienced in mental health and group dynamics, ensuring a safe environment. Through ongoing training, they honed their skills while remaining sensitive to participants' diverse beliefs. Upholding ethical standards and clear communication, we aimed to prevent any adverse effects, fostering trust, respect and well-being throughout the intervention.

Comments 6:  The authors mentioned, "Clergy members have long provided counselling and care to church members...," which, although cited in other literature (see reference [15]), does not resonate with the academic and formal background necessary for this purpose. Specifically, we already know that in such studies, there is a lack of formal education on how to address these aspects in clinical practice (see Naziha S. ; Aleati et al., A. Moreira-Almeida et al.).

Response 6:  We appreciate the reviewer's feedback on the discussion surrounding the role of clergy members in providing counseling and care, particularly in the context of mental health services. While it is acknowledged that clergy members historically have played a significant role in offering support to their communities, we recognize the importance of ensuring a balance between the traditional roles of clergy and the formal education necessary for addressing mental health concerns effectively.

Several studies have highlighted the propensity of individuals to seek help for mental disorders from clergy members over psychiatrists or general medical practitioners, emphasizing the pivotal role clergy play in mental health service delivery (Williams et al., 2014). Recent surveys have also indicated a preference among many individuals, especially women, for receiving therapy from trained clergy members (Iheanacho et al., 2021).

In clinical practice, guidelines often recommend the adoption of healthy lifestyle habits as an initial intervention for conditions such as depression. Should lifestyle changes prove insufficient, generic psychosocial interventions like self-help groups or psychological therapy can be pursued (Malhi et al., 2018).

It is noteworthy that clergy and pastors frequently undergo training in counseling as part of their seminary education, adhering to established standards of practice when providing counseling services (Eliason, 2000; Olliff, 2006). Beyond their spiritual leadership roles, clergy members often serve as mental health gatekeepers within their communities, acting as counselors and referral agents (McDonald, 1984).

Given the ongoing shortage of mental health professionals, the training of lay individuals to deliver mental health interventions in community settings has emerged as a promising strategy to alleviate the burden on the public healthcare system (Iheanacho et al., 2021).

While we uphold the significance of academic and professional qualifications in mental health practice, it is crucial to consider and respect clients' preferences in seeking care. One potential strategy to address this balance is the implementation of a collaborative treatment model that integrates both mental health professionals and clergy members in the treatment process, promoting a holistic approach to care (Breuninger et al., 2014).  Individuals with mild mental health issues can be treated by clergy in the community, but they should refer to mental health professionals if necessary.  During the recruitment and screening for participants, two intended participants were referred to psychiatric consultation and a non-governmental organization (Section 2.4 recruitment and screening for participants) in p. 3 of the manuscript.

We value the reviewer's input on this topic and have added the following in P.2 line 87 under Session 1.1 Spirituality, religion, and depression of the manuscript to ensure a well-rounded exploration of the roles and responsibilities of clergy members in mental health care delivery, while also emphasizing the importance of collaboration and adherence to best practices in mental health interventions:

“Clergy should refer the individual to mental health services if his/her condition has deteriorated or become unmanageable.”

Moving forward, it will be essential to consider implementing training program to equip clergy members with the requisite skills and knowledge before they engage in intervention delivery.  Alternatively, screening their abilities and qualifications could also be a crucial step to guarantee the effectiveness and credibility of the intervention. Adopting a “train the trainers” model can be a valuable approach to enhance the capacity and proficiency of clergy members in delivering interventions. By providing comprehensive training and guidance, we can ensure that they are well-prepared to support participants effectively. Moreover, maintaining strict adherence to the intervention protocol to enhance treatment fidelity will be paramount in future studies. This practice will help uphold the integrity and consistency of the intervention delivery, ultimately contributing to the reliability and validity of the study outcomes.

Comments 7:  The taxonomy of "Forgiveness" [Line 73] in the context of depressive symptoms is highly case-dependent. This needs to be defined on a case-by-case basis and personalized rather than generalized.

Response 7: We appreciate the insightful feedback regarding the taxonomy of "Forgiveness" in relation to depressive symptoms, emphasizing the need for a case-by-case and personalized approach rather than a generalized one.

In response to this valuable input, we have carefully revised the manuscript to reflect the importance of addressing forgiveness within the context of depressive symptoms on an individualized basis. We have emphasized the necessity of tailoring the assessment and intervention processes to suit the unique needs and circumstances of each case, rather than following a one-size-fits-all approach. The revision is made in P. 3 Line 106 under Section 1.2 Therapeutic components of the manuscript:

It is essential that forgiveness should be approached in a personalized manner, taking into account the unique circumstances, beliefs, and experiences of individuals dealing with depressive (Martos Algarra, 2022; Schönherr, 2024).

The updated manuscript now underscores the critical role of personalized care and individualized consideration in understanding and applying the concept of forgiveness in the context of depressive symptoms. By highlighting the significance of a case-specific approach, we aim to enhance the relevance and applicability of the research findings to real-world scenarios where the impact of depressive symptoms varies significantly among individuals.

Moreover, the group facilitators would be sensitive to participants’ needs and address the “individual uniqueness” during the intervention (Supplementary Table S1, under Integration with CBT, positive psychology, and spirituality/religiosity of Forgiveness and freedom).

The following paragraph is added to P.14 line 547 under Section 4. Discussion of the manuscript to highlight the treatment integrity of the group facilitators to ensure an individualized approach in the context of forgiveness:

In our study, we also recognized the potential negative impacts of spirituality and religiosity, and the uniqueness of each individual. To safeguard participants, our carefully selected facilitators underwent comprehensive training and adhered strictly to ethical guidelines. They were experienced in mental health and group dynamics, ensuring a safe environment. Through ongoing training, they honed their skills while remaining sensitive to participants' diverse beliefs. Upholding ethical standards and clear communication, we aimed to prevent any adverse effects, fostering trust, respect and well-being throughout the intervention.

Comments 8:  Did the authors have any criteria for the "social identity" of the participants as a key element of such therapies? If so, how and which method was implemented?

Response 8: Thank you for your question. The study did not explicitly impose specific criteria for social identity as a prerequisite for participation, except for the inclusion of adult aged 18-24 and Hong Kong Chinese resident who can communicate in Cantonese. The study did not restrict participation based on social identity factors such as religious beliefs. Both Christians and non-religious individuals were eligible to participate in the intervention. By maintaining an inclusive approach to participant selection, the study aimed to explore the effects of the intervention across diverse social identities without imposing restrictions based on specific criteria related to social identity.

                The inclusion criteria of adults aged 18-64 and Hong Kong Chinese residents who can communicate in Cantonese indirectly address certain aspects of social identity.

  1. Age Group Inclusion (18-64): It was intended to exclude adolescent and elderly participants as their clinical presentation and response to treatment can be different from adult participants (Malhi & Mann, 2018).  Different age groups often have distinct social norms, roles, and experiences that can shape individual identities and interactions. The adult group encompassing various life stages and experiences.
  2. Cultural Group (Hong Kong Chinese residents who can communicate in Cantonese): Focusing on Hong Kong Chinese residents who can communicate in Cantonese acknowledges the importance of cultural identity as a key element of social identity. It is crucial to consider clients’ cultural and linguistic context in psychotherapy (Bernal et al., 2009). Shared cultural background, language, and heritage can influence how individuals perceive themselves within the social context.

While the study did not have explicit criteria for social identity beyond age and cultural background, the chosen inclusion criteria indirectly capture certain aspects of social identity related to age group and cultural affiliation. These criteria help ensure a level of homogeneity within the study sample based on specific social characteristics, allowing for a more targeted investigation of the intervention's effects within these defined social groups.

Comments 9:  The authors mentioned the required number (through power analysis) to obtain the necessary power. However, it appears to me that the number of participants is insufficient. If I am mistaken, please let me know; otherwise, how do the authors justify the lack of sufficient power?

Response 9: Thank you for your raising the issue of sample size inadequacy in our study. Your feedback has prompted us to reevaluate the study's design and considerations for future research. We appreciate you bringing this important issue to our attention. Your insightful observation has led us to reconsider our approach and reflect on the implications of this discrepancy. Although the calculated sample size was 58 based on a conservative effect size of d=0.38 and a statistical power of 0.8, our study included 57 participants, slightly below this target.

Despite the shortfall in participant numbers, a post hoc power analysis, detailed in Appendix I, revealed promising results. The observed effect sizes (Cohen’s d) for the between group comparison ranged from 0.353 to 1.801, with most of outcome variables exceeded 0.5 indicating a moderate to strong effect. Similarly, the observed effect sizes for the within group comparison also showed moderate to strong effect for most of the outcome variable. Additionally, the calculated power for around half of the outcome measures exceeded 0.8, suggesting robust statistical power in detecting these effects.

                We acknowledge the importance of statistical power in drawing reliable conclusions. Despite the sample size being slightly below the calculated figure, the observed effect sizes and high power values provide confidence in the study's results.

                Your feedback has been instrumental in our reassessment of the study's design and considerations for future research. We appreciate your engagement and valuable insights in enhancing the quality of our work.

Comments 10:  With 8 out of 57 participants declaring different religions for such treatment, do the authors not think this may impact the results?

Response 10: Thank you for your thoughtful feedback regarding the potential impact of participants' diverse religious backgrounds on the outcomes of our study. We appreciate the opportunity to address this important point raised during the review process.

In our study, we acknowledge and respect the diversity of religious backgrounds among our participants. While the intervention was rooted in Protestant Christianity, it was intentionally designed to be inclusive and welcoming to individuals of all faiths and belief systems. Participants from various religious backgrounds, including Protestant Christians, Roman Catholics, and those who declared themselves as non-religious, were included in the study.

Some of the themes in the spiritual intervention such as connectedness, forgiveness, hope, gratitude, and relapse prevention, which are relevant and beneficial across different spiritual beliefs and cultural practices. We ensured that the program did not impose specific religious rituals or beliefs that might be exclusive to certain faith traditions.

Participants with diverse spiritual beliefs and cultural practices were treated with respect and sensitivity throughout the intervention. By fostering an environment of inclusivity and openness, we aimed to provide a supportive space where individuals from various religious backgrounds could engage with the intervention content and benefit from its therapeutic aspects.

We believe that the fundamental themes and core principles of the intervention went beyond particular religious associations, enabling individuals with various spiritual beliefs to discover significance and benefit from the program. The positive outcomes observed in our study, including significant improvements in depressive symptoms, anxiety, spiritual experience, hope, self-esteem, and social support, underscore the effectiveness of the intervention in a context that embraces diversity and inclusivity.

Additionally, we explored the moderating effects of religious backgrounds on effect of intervention on different grouping (religious group vs non-religious group), finding that differences in religious affiliation did not significantly influence the intervention outcomes. This suggests that the intervention's effectiveness was consistent across participants with diverse religious backgrounds.

Comments 11:  Given the variety of activities in your treatment, have you tried to check which factor in terms of the participants had more impact on them or not?

Response 11: Thank you for the insightful feedback provided by the reviewer. While our quantitative analysis focuses on the overall effectiveness of our intervention, we acknowledge the importance of understanding specific factors' impacts on participants.

To explore further, we conducted qualitative evaluations through focus groups following the intervention. These focus groups aimed to explore deeper into participants' perspectives on the role of spirituality in treating depression, identify the most beneficial activities, and explain the specific therapeutic components that were most impactful.

While the quantitative analysis demonstrates the effectiveness of the intervention in improving various outcomes, the qualitative evaluation allows us to capture the experiences and insights of participants. By exploring participants' perspectives through qualitative methods, we aim to gain a richer understanding of how the intervention influenced their well-being beyond statistical measures.

We are currently in the process of analyzing the qualitative data gathered from the focus groups. The insights gained from this analysis will provide a more comprehensive view of the intervention's impact from the participants' standpoint. We intend to incorporate these qualitative findings into our study to offer a more holistic understanding of the intervention's effectiveness.

While our current approach may not pinpoint individual factors' influence, we are dedicated to evaluating the intervention as a whole. Our qualitative findings will be presented in a separate paper, enriching our understanding of participants' experiences and perspectives.

Comments 12:  Taking into account your Table S2, why did you not give your participants more options?

Response 12: We appreciate the reviewer's attention to the details of Table S2 and the concern regarding the number of options provided in the checklist. We acknowledge the importance of clarity in the checklist for interventionists' self-monitoring during treatment sessions.

We would like to clarify Table S2 (revised as Table S3) is the checklist for the implementation fidelity. The limited options of "always/almost always," "sometimes," and "missed opportunity" in Table S2 (revised as Table S3) were intentionally chosen to focus on key criteria essential for evaluating the quality of the interventionist's performance. These options were designed to capture the frequency with which the interventionist met the specified criteria, allowing for a clear and concise assessment of their adherence to the intervention protocol.

While providing additional options could offer more precise feedback, we aimed to streamline the checklist to prioritize essential aspects of the intervention process. The simplicity of the response choices helps maintain the checklist's efficiency and facilitates consistent and straightforward self-evaluation by interventionists.

We believe that the current structure of Table S2 (revised as Table S3) effectively supports the intended purpose of enabling interventionists to monitor their performance during treatment sessions and ensuring adherence to the intervention protocol. However, we will consider the reviewer's feedback for future revisions and explore potential refinements that could enhance the checklist's comprehensiveness without compromising its clarity and utility.

Thank you for the valuable feedback, and we are committed to continuously improving the quality and effectiveness of our intervention monitoring processes based on constructive input from reviewers and stakeholders.

Comments 13:  You have some participants who did not attend all sessions (right?). Did you use their data also in the analysis? Or not?

Response 13: Thank you for highlighting the issue regarding participants who did not attend any session during the study. I appreciate the opportunity to clarify the approach taken regarding their data in the analysis.

In our study, we encountered 2 participants who did not attend any session after their completion of the baseline questionnaire.  As outlined in section 2.13 Statistical Analysis in P. 7 line 308 of the manuscript, "The study used an intention-to-treat (ITT) strategy, and any missing values in the outcome variables were replaced with the corresponding baseline values". This approach is a common method in clinical trials, aiming to maintain the original randomization order and include all participants in the analysis according to their assigned groups, regardless of receiving the complete treatment, only partly or nothing at all (Twisk et al., 2020). This approach was chosen to prevent biases that could arise from excluding participants who did not attend all sessions, ensuring a more conservative and inclusive analysis that aligns with the initial study design and randomization process. 

Furthermore, sensitivity is performed to assess the robustness of the study results and strengthen the credibility of the findings (Thabane et al., 2013).  Please refer to section 3.8 Sensitivity analysis in P. 13 line 447, “A sensitivity analysis was conducted to compare the results of the ITT analysis and per-protocol (PP) analysis.”

Comments 14:  Did you consider the factor of age? Are there any differences by age?

Response 14: Thank you for your insightful question regarding the consideration of age as a factor in our study. We did indeed explore the impact of age as a moderator in our analysis. The results in Section 3.6 Moderation effect and Table 4 in p 11 of the manuscript indicated significant differences by age in the effects of the spiritual connectivity intervention on various outcome variables.

Specifically, our findings revealed that age moderated the intervention effects on depression, hope, agency thinking, perceived social support, and support from friends. Older participants demonstrated a more pronounced positive response to the intervention compared to their younger counterparts. This aligns with existing research that suggests age can influence the outcomes of spiritual programs ((Montes & Tonigan, 2017; Stearns et al., 2018).     

These results highlight the importance of considering age as a relevant factor in the design and implementation of interventions like the one studied. Moving forward, we acknowledge the need to adapt our program to better cater to the needs of younger adults, ensuring that they can also derive significant benefits from the intervention. Future investigations could delve deeper into the underlying factors contributing to the age-related differences observed, potentially focusing on aspects such as participants' expectations of the intervention at the recruitment stage. This understanding will be instrumental in refining the spirituality program to suit the diverse needs and preferences of participants across different age groups, thereby enhancing the overall impact and effectiveness of the intervention. This is also included in the p. 14 line 533 under Section 4 Discussion of the manuscript.

Comments on the Quality of English Language

Minor edits are needed, and transitions between some sentences are not clear. Overall English level is good, but some minor are there.

Thanks for reviewer’s comment and we appreciate your suggestion that minor editing of English is required. However, we would like to let you know that we have already undergone English language editing by a professional editor, MDPI.

We understand that there may be different perception on the clarity, we are confident in the quality of the editing service done by MDPI. Nevertheless, we have reviewed the manuscript again and sought for second opinion from a native English speaker to review our manuscript to ensure the standard requirement of clarity. Minor revisions on English language are made in the manuscript, which is highlighted in blue.

We hope the above points can clarify our intentions for this study. Thank you again for your feedback and for taking time to review my work.

References:

Bernal, G., Jiménez-Chafey, M. I., & Domenech Rodríguez, M. M. (2009). Cultural adaptation of treatments: A resource for considering culture in evidence-based practice. Professional Psychology: Research and Practice, 40(4), 361.

Breuninger, M., Dolan, S. L., Padilla, J. I., & Stanford, M. S. (2014). Psychologists and clergy working together: A collaborative treatment approach for religious clients. Journal of Spirituality in Mental Health, 16(3), 149-170.

Eliason, G. T. (2000). The credentials and practices of Presbyterian clergy in pastoral counseling. Duquesne University.

Fallot, R. D. (2001). Spirituality and religion in psychiatric rehabilitation and recovery from mental illness. International Review of Psychiatry, 2001, Vol.13(2), p.110-116, 13(2), 110-116. https://doi.org/10.1080/09540260120037344

Gomi, S., Starnino, V., & Canda, E. (2014). Spiritual Assessment in Mental Health Recovery. Community Ment Health J, 50(4), 447-453. https://doi.org/10.1007/s10597-013-9653-z

Iheanacho, T., Nduanya, U. C., Slinkard, S., Ogidi, A. G., Patel, D., Itanyi, I. U., Naeem, F., Spiegelman, D., & Ezeanolue, E. E. (2021). Utilizing a church-based platform for mental health interventions: Exploring the role of the clergy and the treatment preference of women with depression. Global Mental Health, 8, e5.

Kao, L. E., Peteet, J. R., & Cook, C. C. H. (2020). Spirituality and mental health. Journal for the study of spirituality, 10(1), 42-54. https://doi.org/10.1080/20440243.2020.1726048

Kessler, R. C., & Bromet, E. J. (2013). The Epidemiology of Depression Across Cultures. Annu. Rev. Public Health, 34(1), 119-138. https://doi.org/10.1146/annurev-publhealth-031912-114409

Li, W., Denson, L. A., & Dorstyn, D. S. (2018). Understanding Australian university students’ mental health help‐seeking: An empirical and theoretical investigation. Australian Journal of Psychology, 70(1), 30-40. https://doi.org/10.1111/ajpy.12157

Malhi, G. S., & Mann, J. J. (2018). Depression. The Lancet, 392(10161), 2299-2312. https://doi.org/10.1016/S0140-6736(18)31948-2

Malhi, G. S., Outhred, T., Hamilton, A., Boyce, P. M., Bryant, R., Fitzgerald, P. B., Lyndon, B., Mulder, R., Murray, G., Porter, R. J., Singh, A. B., & Fritz, K. (2018). Royal Australian and New Zealand College of Psychiatrists clinical practice guidelines for mood disorders: major depression summary. Medical journal of Australia, 208(4), 175-180. https://doi.org/10.5694/mja17.00659

Martos Algarra, C. (2022). The role of forgiveness in disclosure and victim suport after a patient safety incident.

McDonald, M. J. (1984). CLERGY AS GATEKEEPERS: NETWORKS OF COUNSELING AND REFERRAL (CAREGIVERS, PASTORAL COUNSELING, RELIGION). Purdue University.

Montes, K. S., & Tonigan, J. S. (2017). Does age moderate the effect of spirituality/religiousness in accounting for Alcoholics Anonymous benefit? Alcoholism treatment quarterly, 35(2), 96-112.

Moreira-Almeida, A., & Bhugra, D. (2021). Religion, spirituality, and mental health: setting the scene. Spirituality and mental health across cultures, 11-26.

Olliff, K. L. (2006). Assessing the comfort level of clergy in the use of counseling skills Capella University].

Schönherr, J. (2024). Forgiveness: Overcoming versus Forswearing Blame. Journal of Applied Philosophy, 41(1), 66-84.

Smith, T. B., McCullough, M. E., & Poll, J. (2003). Religiousness and Depression: Evidence for a Main Effect and the Moderating Influence of Stressful Life Events. Psychol Bull, 129(4), 614-636. https://doi.org/10.1037/0033-2909.129.4.614

Stearns, M., Nadorff, D. K., Lantz, E. D., & McKay, I. T. (2018). Religiosity and depressive symptoms in older adults compared to younger adults: Moderation by age. J Affect Disord, 238, 522-525.

Thabane, L., Mbuagbaw, L., Zhang, S., Samaan, Z., Marcucci, M., Ye, C., Thabane, M., Giangregorio, L., Dennis, B., Kosa, D., Borg Debono, V., Dillenburg, R., Fruci, V., Bawor, M., Lee, J., Wells, G., & Goldsmith, C. H. (2013). A tutorial on sensitivity analyses in clinical trials: the what, why, when and how. BMC medical research methodology, 13(1), 92-92. https://doi.org/10.1186/1471-2288-13-92

Twisk, J. W., Rijnhart, J. J., Hoekstra, T., Schuster, N. A., Ter Wee, M. M., & Heymans, M. W. (2020). Intention-to-treat analysis when only a baseline value is available. Contemporary clinical trials communications, 20, 100684.

Williams, L., Gorman, R., & Hankerson, S. (2014). Implementing a Mental Health Ministry Committee in Faith-Based Organizations: The Promoting Emotional Wellness and Spirituality Program. Soc Work Health Care, 53(4), 414-434. https://doi.org/10.1080/00981389.2014.880391

Appendix I

Post hoc power analysis using the effect sizes and sample sizes

Primary outcomes

Effect Size d (Between Group )

Effect Size f

Power (1 - β)

Depression (PHQ9)

-1.801

-0.901

0.9999987

Anxiety (GAD7)

-1.605

-0.803

0.9999644

Secondary outcomes

Spiritual experience (DSES)

0.879

0.440

0.9017955

Hope (SHS)

1.298

0.649

0.9977112

Agency Thinking (SHS-Agency)

1.424

0.712

0.9995174

Pathway Thinking (SHS-Pathway)

0.960

0.480

0.9437788

Meaning in Life (MLQ)

0.563

0.282

0.5495748

Presence of Meaning in Life (MLQ-Presence)

0.585

0.293

0.5815465

Search of Meaning in Life (MLQ-Search)

0.353

0.177

0.7418757

Self-Esteem (RSES)

0.654

0.327

0.6761813

Perceived Social Support (MSPSS)

0.508

0.254

0.4672302

Perceived support from family (MSPSS-family)

0.471

0.236

0.4148289

Perceived support from friends (MSPSS-friend)

0.386

0.193

0.2972984

Perceived support from significant others (MSPSS-Sig.)

0.420

0.210

0.3420189

Primary outcomes

Effect Size d (Within Group at T1 )

Effect Size f

Power (1 - β)

Depression (PHQ9)

-1.452

-0.726

1.0000000

Anxiety (GAD7)

-1.124

-0.562

1.0000000

Secondary outcomes

Spiritual experience (DSES)

0.308

0.154

0.6272848

Hope (SHS)

0.633

0.317

0.9969426

Agency Thinking (SHS-Agency)

0.702

0.351

0.9994139

Pathway Thinking (SHS-Pathway)

0.479

0.240

0.9994139

Meaning in Life (MLQ)

0.232

0.116

0.4056727

Presence of Meaning in Life (MLQ-Presence)

0.338

0.169

0.7078522

Search of Meaning in Life (MLQ-Search)

0.043

0.022

0.0622941

Self-Esteem (RSES)

0.503

0.252

0.9622906

Perceived Social Support (MSPSS)

0.515

0.258

0.9690401

Perceived support from family (MSPSS-family)

0.324

0.162

0.6712293

Perceived support from friends (MSPSS-friend)

0.540

0.270

0.979562

Perceived support from significant others (MSPSS-Sig.)

0.499

0.250

0.959790

Reviewer 2 Report

Comments and Suggestions for Authors

- Rewrite the introduction to emphasize the unique contributions of this work and how it varies from other studies on spiritual therapies for depression. Stress the novelty of the Spiritual Connectivity Intervention (SCI) approach

- Clarify the unique contributions of this study in the introduction and how it differs from existing research on spiritual interventions for depression.

- Provide detailed descriptions of the datasets used, including characteristics and sample sizes, with access links for reproducibility.

- Include the code used for experimental implementations as an appendix to enhance transparency and reproducibility.

- Expand on the SCI methodological details, illustrating the integration of cognitive behavioral therapy and positive psychology principles with diagrams.

- Provide specific examples and case studies to demonstrate the practical implications of the performance metrics used.

- Discuss ethical considerations and potential misuse of spiritual interventions in further detail, along with advice for mitigating negative consequences.

- Please ensure that every figure and table has the proper labels, is well resolved, high resolution and quality.

- Include recent studies in the literature review and critically compare methodologies, findings, and limitations of different spiritual interventions for depression.

- Provide further details about the significance of the findings related to spiritual experience, anxiety, and depression, as well as potential directions for future research.

- Provide a detailed discussion of the results, comparing with state-of-the-art methods, and highlight strengths and weaknesses of the SCI method.

- Explain about how the SCI might be approached to other mental health issues and cultural contexts.

Comments on the Quality of English Language

 Minor editing of English language required

Author Response

We would like to express our sincere gratitude to reviewer for your thorough reading of this manuscript and for the thoughtful and constructive comments, which helped improve the quality of the paper. We appreciate the time and effort that you dedicated to providing feedback on it. After carefully addressing all the comments, the refinements are made in the revised manuscript, which is highlighted in red.  The detailed responses are summarized in Point-by-point response to Comments and Suggestions for Authors.

Point-by-point response to Comments and Suggestions for Authors

Comments 1: Rewrite the introduction to emphasize the unique contributions of this work and how it varies from other studies on spiritual therapies for depression. Stress the novelty of the Spiritual Connectivity Intervention (SCI) approach.

Response 1: Thank you for your feedback and we understand your expectations. The SCI is novel in its focus on Christian faith and connectedness. It can also fill the service gap for this difficult-to-reach population. Revision is made in the introduction in P.1 and 2 line 33 and 45 respectively under Section 1 Introduction of the manuscript:

“Depression is one of the most under-reported and under-treated mental health issue (Kessler & Bromet, 2013). This may be due to ignorant or lack of knowledge about their health problem. People’s belief in mental health or stigmatization might also hinder their help-seeking behavior (Li et al., 2018).

                What distinguishes our study is the explicit focus on fostering connectedness, a vital component that sets it apart from other interventions in this domain. By implementing a community-oriented approach, we aim to bridge the existing service gap and extend support to a segment of the population that is often hard to reach. This initiative is particularly significant as it aims to facilitate early identification and intervention for individuals who might otherwise lack access to or engagement with conventional mental health services, potentially alleviating the strain on the public healthcare system”

The revised introduction now highlights that depression is a significantly under-reported and under-treated mental health concern, potentially due to a lack of awareness about the condition and the impact of societal beliefs and stigma on help-seeking behaviors. Additionally, the updated introduction places specific emphasis on the distinct focus of the study, which lies in fostering connectedness as a pivotal element that sets it apart from other interventions in the field. This restructuring aims to underscore the unique contributions and innovative approach of the study in addressing the service gap for individuals with depressive symptoms.

Comments 2:  Clarify the unique contributions of this study in the introduction and how it differs from existing research on spiritual interventions for depression.

Response 2: Thank you for your feedback and comments on our study. We appreciate your concern on the distinctive contribution of our research. Our study introduces a novel approach that centers on Christian faith and connectedness to address depression through the SCI, setting it apart from existing research on spiritual interventions for depression. Please refer to the response to comment 1 above and the revision made in P.1 and 2 line 33 and 45 respectively under Section 1 Introduction of the manuscript.

    Some recent literatures are reviewed and their methodologies, findings, and limitations are discussed and in P.2 line 71 under Section 1.1 Spirituality, religion and depression of the manuscript:

“Portnoff et al. (2017) highlighted that heightened spirituality correlated with reduced depression risk across diverse cultural settings, yet the heavy reliance on self-reported data raises concerns about potential biases. Anderson et al. (2015) demonstrated the efficacy of faith-adapted cognitive behavior therapy (CBT) in treating depression, outperforming standard CBT and control conditions with a moderate effect size ranging from .31 to .59. However, the lack of detailed randomization descriptions and researcher allegiance in these trials introduces potential biases. Similarly, Captari et al. (2018) showcased the benefits of religious/spiritual (R/S) adapted psychotherapy over standard treatment for depression with an effect size of Hedges’ g = .33. Nonetheless, they noted the absence of proper comparative secular and R/S treatments in many studies, which impacts the interpretation of results.”

Furthermore, it is mentioned in P.3 line 125 under Session1.2 Therapeutic components “there is a dearth of studies using rigorous methodological designs, such as randomized controlled trials (RCTs), to evaluate the efficacy of spiritual interventions for individuals with depression.”

Unlike many previous studies that lack proper comparative secular and religious/spiritual treatments, our research rigorously incorporates these control groups to provide a comprehensive evaluation of the SCI's effectiveness. This methodological rigor not only enhances the robustness of our findings but also enables a clearer interpretation of the intervention's unique impact on depressive symptoms. By addressing this gap in the literature, our study aims to contribute valuable insights into the efficacy of spiritual interventions for individuals experiencing depression, paving the way for more sophisticated and tailored treatment approaches in the field. We believe that our study adds value and unique insight to the current literature in this field.

Comments 3:  Provide detailed descriptions of the datasets used, including characteristics and sample sizes, with access links for reproducibility.

Response 3: Thank you for your insightful feedback regarding the datasets used in our study. We have taken great care in preparing these datasets to facilitate reproducibility in research.

In the interest of transparency and reproducibility, we have made the datasets available for this review. Below is the link where you can access and download the datasets:

We acknowledge the importance of enhancing reproducibility in research. However, we encountered a challenge regarding the public availability of the data due to our initial consent process with participants. During the consent process prior to the intervention, participants were not informed that their data would be made publicly available. Our ethical commitment at that time was to ensure the data's confidentiality and eventual destruction upon the study's completion.

Given this constraint, we are unable to seek additional consent from participants for public data sharing. Moving forward, we will address this issue by explicitly mentioning it in our information sheet and consent form for future studies to ensure transparency and participant understanding.

The dataset is available for your review, and we are open to providing access upon request for verification purpose or collaboration. We are dedicated to upholding ethical research practices while promoting collaboration and transparency in our work.

Comments 4:  Include the code used for experimental implementations as an appendix to enhance transparency and reproducibility.

Response 4: We appreciate your feedback and your emphasis on transparency and reproducibility in research. In response to your suggestion, we have diligently prepared the SPSS syntax used for the experimental implementations. This code is included in the Appendix I of supplementary materials to enhance the transparency of our study. 

The SPSS syntax provides a detailed account of the procedures and analyses conducted during the experiment, allowing for a clear understanding of the steps taken and the statistical methods applied. By including this code, we aim to facilitate the replication of our study and promote a transparent research process.

Comments 5:  Expand on the SCI methodological details, illustrating the integration of cognitive behavioral therapy and positive psychology principles with diagrams.

Response 5: Thank you for your comment and suggestion. A simple diagram (Figure 1) based on the work of Leung and Li (2023) is added in P. 6 line 252 under Section 2.7 Interventions of the manuscript:

                 Figure 1. Spiritual connectivity intervention framework

                Furthermore, a concept mapping table is added in the Supplementary Material file with the left column outlines the content of each session in the SCI while the right column specifies how cognitive behavioral therapy (CBT), positive psychology principles, and spirituality/religiosity in group work can be integrated into each session to enhance the therapeutic process and outcomes. The following is added in P.5 line 225 under Section 2.7 of the manuscript:

The integration of spirituality/religiosity in the group therapy approach alongside cognitive behavioral therapy and positive psychology principles is implemented in every session (Supplementary File, Table S1 and S2).

Please refer to the Appendix I for details of concept mapping on integration of treatment strategies.

Comments 6:  Provide specific examples and case studies to demonstrate the practical implications of the performance metrics used.

Response 6: Thank you for your feedback regarding the request for specific examples and case studies to illustrate the practical implications of the performance metrics used in our study.

Our research is grounded in a randomized controlled study design, focusing on quantitative analyses and the evaluation of intervention outcomes. While we do not feature qualitative case studies, we have employed robust quantitative methods to assess the effectiveness of the intervention.

As reduction of 5 points in the Patient Health Questionnaire-9 (PHQ-9) score post-intervention is recognized as reaching clinical significance (Kroenke et al., 2020; Löwe et al., 2004; Wong et al., 2021). In line with this criterion, we have utilized this benchmark to evaluate the impact of our intervention on depressive symptoms. Furthermore, we have reported the percentage of participants who achieved this clinically significant reduction in the PHQ-9 score in P.13 line 453 under Section 3.9 Clinical Significance of the manuscript.

By adhering to established benchmarks and reporting metrics such as the percentage of participants meeting clinical significance thresholds, we aim to provide a clear and clinically meaningful interpretation of our study results. These quantitative outcomes serve as essential indicators of the intervention's effectiveness in addressing depressive symptoms within our study population.

Comments 7:  Discuss ethical considerations and potential misuse of spiritual interventions in further detail, along with advice for mitigating negative consequences.

Response 7: Thank you for reviewer’s thoughtful comments on the ethical considerations and potential misuse of spiritual interventions in research, careful examination of these aspects are crucial in ensuring the integrity and ethical conduct of the study.

Ethical considerations, including informed consent, confidentiality, cultural sensitivity, competence, and beneficence, are fundamental in safeguarding participants' rights and well-being when integrating spiritual interventions. Informed consent was provided to ensure participants fully understood the nature of the spiritual interventions, including potential risks and benefits, and their rights to withdraw at any stage. Confidentiality was maintained, with all participants remaining anonymous and individual respondents not being identifiable (The information sheet and consent from are attached for your information, please refer to Appendix II & III). Despite the spiritual connectivity intervention (SCI) being rooted in Christianity, participants’ diverse spiritual belief and cultural practices were respected. The SCI in this study was conducted by three facilitators, with all intervention groups led by the one of the authors, an experienced psychiatric nurse. The waitlist control groups were led by an experienced social worker and another experienced social worker. All three facilitators got extensive experience in serving clients with mental health issues and leading therapeutic groups. The two facilitators of the waitlist control groups received training on spiritual interventions from one of the authors who developed the SCI program to ensure ethical practice and minimize harm.  All facilitators are competent in group counselling. As they are health professionals, they are conscious and committed to benefiting participants and avoiding harm at all times.

                Spiritual intervention, much like any forms of therapy, may entail the risk of causing harm if it is poorly administered (Taylor, 2023).  We acknowledge the importance of addressing the issue, especially concerning clergy members and participants, to prevent any potential misuse of misunderstanding in developing further relationships. The potential misuse of spiritual intervention can lead to spiritual abuse, involving emotional and psychological harm inflicted through coercive and controlling actions within a religious framework or justified by spiritual beliefs (Oakley et al., 2024). If an individual is struggling with guilt or self-blame, a poorly handled focus on forgiveness could potentially exacerbate feelings of alienation and spiritual distress.

                To mitigate these risks, facilitators should strictly adhere to ethical guidelines and receive training on spiritual intervention. They should approach each individual with respect, sensitivity, and an open mind, and non-judgmental to participants’ diverse cultural and spiritual. It is also crucial to monitor the individual’s response to the intervention and adjust the approach as needed. Screening and approval by ethics committee, as well as informed consent are key ethical consideration. Potential participants should be provided with comprehensive information, the opportunity to ask questions, and the freedom to withdraw from the intervention at any time, will be paramount in our future research endeavors. Finally, spiritual intervention should be used in conjunction with other therapeutic approaches as appropriate and it is not a replacement for the conventional treatment.

Comments 8:  Please ensure that every figure and table has the proper labels, is well resolved, high resolution and quality.

Response 8: Thank you for your feedback and attention to detail regarding the figures and tables in our manuscript. We have taken your comments into consideration and have made sure that each figure and table is appropriately labeled, well-resolved, and of high resolution and quality.

By ensuring that all visual elements are clearly presented and meet the required standards, we aim to enhance the readability and comprehensibility of our findings for the readers. Should you have any further suggestions or specific areas of concern regarding the figures and tables, please do not hesitate to let us know so that we can address them promptly.

We appreciate your valuable input and are committed to maintaining the quality and clarity of our visual representations to support the overall integrity of our research. Thank you for your attention to this important aspect of our manuscript.

Comments 9:   Include recent studies in the literature review and critically compare methodologies, findings, and limitations of different spiritual interventions for depression.

Response 9: In response to the reviewer's comments on recent studies examining spiritual interventions for depression, the following studies are included in the literature review in P.2 line 71 under Section 1.1 Spirituality, religion and depression of the manuscript:

Portnoff et al. (2017) highlighted that heightened spirituality correlated with reduced depression risk across diverse cultural settings, yet the heavy reliance on self-reported data raises concerns about potential biases. Anderson et al. (2015) demonstrated the efficacy of faith-adapted cognitive behavior therapy (CBT) in treating depression, outperforming standard CBT and control conditions with a moderate effect size ranging from .31 to .59. However, the lack of detailed randomization descriptions and researcher allegiance in these trials introduces potential biases. Similarly, Captari et al. (2018) showcased the benefits of religious/spiritual (R/S) adapted psychotherapy over standard treatment for depression with an effect size of Hedges’ g = .33. Nonetheless, they noted the absence of proper comparative secular and R/S treatments in many studies, which impacts the interpretation of results.

Comments 10:   Provide further details about the significance of the findings related to spiritual experience, anxiety, and depression, as well as potential directions for future research.

Response 10: Our study’s results indicating a significant increase in spiritual experience alongside a decrease in depressive symptoms and anxiety levels after the SCI. The observed relationship suggests that enhancement in spiritual experience may have a positive impact on mental health outcomes, specifically reducing depressive symptoms and anxiety levels. Moving forward, long-term studies could reveal the enduring impact of SCI on mental health results as time progresses.

The following paragraph is added in P. 13 line 486 under 4. Discussion of the manuscript:

        Participants in both the SCG and WLG reported a significant increase in spiritual experience after participating in the spiritual connectivity program. Previous studies of spiritual intervention on clients with cancer also showed spiritual growth and improved spiritual well-being after the intervention (Afrasiabifar et al., 2021; Nasution et al., 2020; Post et al., 2020). The observed relationships suggests that in increase in spiritual engagement could potentially benefit mental health by notably decreasing symptoms of depression and anxiety. Looking ahead, future research endeavors could delve into the enduring effects of the SCI on mental health outcomes over an extended period. Longitudinal studies have the potential to elucidate the sustained impact of the intervention on aspects like depression and anxiety, offering insights into the long-term efficacy of SCI in fostering improved mental well-being.

Comments 11:  Provide a detailed discussion of the results, comparing with state-of-the-art methods, and highlight strengths and weaknesses of the SCI method.

Response 11: Thank you for reviewer’s comment.  The present examined the impact of the SCI on various psychological outcomes in adults with mild to moderate depressive symptoms.  The findings indicate that the SCI led to significant improvements in depressive symptoms, anxiety levels, spiritual experiences, hope, self-esteem, and perceived social support in the intervention group compared to the waitlist control group. These positive effects were maintained at the follow-up measurement, particularly for depressive symptoms and anxiety levels. These has been discussed in the manuscript in details. Some revision has been made in P.14 line 534 and P.15 line 555 under Section 4. Discussion of the manuscript:

       Regarding the sensitivity analysis, the results indicated that the statistical significance of the outcome measures did not differ between the intention-to-treat (ITT) analysis and the per-protocol (PP) analysis. This suggests that the findings are robust and not heavily influenced by dropouts or non-compliance with the program. Both analysis methods yielded consistent results, supporting the validity of the study's findings.

The strength of this study lies in its demonstration of the broad efficacy in enhancing diverse psychological outcomes, indicating its comprehensive approach to mental health.  The findings suggest that the intervention may work by cultivating a sense of meaning and connection to a higher power/god, rather than through social support mechanisms. The identification of age as a moderating factor provides valuable insights for tailoring the intervention to different age groups effectively. A significant proportion of SCI participants achieved substantial reductions in depressive symptoms, highlighting the interventions’ practical relevance.

The weaknesses of the study has been discussed in details in P15 line 563 under Section 4.1 Limitations of the manuscript.

Comments 12:   Explain about how the SCI might be approached to other mental health issues and cultural contexts.

Response 12: Thank you for your insightful comment. The application of the SCI could potentially extend to other mental health issues and cultural contexts. In terms of other mental health issues, the components of the SCI such as divine connection, forgiveness and freedom, suffering and transcendence, hope, gratitude, and relapse prevention could be beneficial for individuals dealing with a variety of mental health disorders. For instance, the focus on forgiveness and freedom could be beneficial for individuals dealing with trauma or post-traumatic stress disorders, while focus on hope and gratitude could be beneficial for individuals dealing with anxiety disorder. However, further research would be needed to confirm these hypotheses and to adapt the SCI to these contexts. In terms of cultural contexts, the SCI could potentially be adapted to different religious or spiritual traditions. For instance, the concept of divine connection could be interpreted in different ways depending on the individual’s religious or spiritual beliefs.  Similarly, the concepts of forgiveness, suffering, hope, and gratitude are universal and could be interpreted in different ways depending on the cultural context. However, it would be important to work closely with individuals from these cultural contexts to ensure that the SCI is culturally sensitive and appropriate. We appreciate your suggestion and have incorporated these points into P. 15, line 582 of Section 4.1 Implications of the manuscript:

“SCI shows potential for application across diverse mental health issues, with components like forgiveness and freedom holding relevance for trauma and post-traumatic stress disorders, while a focus on hope and gratitude could be beneficial for individuals with anxiety disorder. SCI could be potentially adapted to different religious or spiritual traditions with those universal themes of forgiveness, suffering, hope, and gratitude. Health care professionals need to be flexible and respect clients’ cultural needs in implementing the intervention.”

  1. Response to Comments on the Quality of English Language

Point 1: Minor editing of English language required.

Response 1: Thanks for reviewer’s comment and we appreciate your suggestion that minor editing of English is required. However, we would like to let you know that we have already undergone English language editing by a professional editor, MDPI.

We understand that there may be different perception on the clarity, we are confident in the quality of the editing service done by MDPI. Nevertheless, we have reviewed the manuscript again and sought for second opinion from a native English speaker to review our manuscript to ensure the standard requirement of clarity. Minor revisions on English language are made in the manuscript, which is highlighted in blue.

We hope the above points can clarify our intentions for this study. Thank you again for your feedback and for taking time to review my work.

References:

Afrasiabifar, A., Mosavi, A., Jahromi, A. T., & Hosseini, N. (2021). A Randomized Controlled Trial Study of the Impact of a Spiritual Intervention on Hope and Spiritual Well-Being of Persons with Cancer. Investigación y Educación en Enfermería, 39(3). https://doi.org/10.17533/udea.iee.v39n3e08

Anderson, N., Heywood-Everett, S., Siddiqi, N., Wright, J., Meredith, J., & McMillan, D. (2015). Faith-adapted psychological therapies for depression and anxiety: Systematic review and meta-analysis. J Affect Disord, 176, 183-196. https://doi.org/10.1016/j.jad.2015.01.019

Captari, L. E., Hook, J. N., Hoyt, W., Davis, D. E., McElroy‐Heltzel, S. E., & Worthington, E. L. (2018). Integrating clients’ religion and spirituality within psychotherapy: A comprehensive meta‐analysis. Journal of Clinical Psychology, 74(11), 1938-1951. https://doi.org/10.1002/jclp.22681

Kessler, R. C., & Bromet, E. J. (2013). The Epidemiology of Depression Across Cultures. Annu. Rev. Public Health, 34(1), 119-138. https://doi.org/10.1146/annurev-publhealth-031912-114409

Kroenke, K., Stump, T. E., Chen, C. X., Kean, J., Bair, M. J., Damush, T. M., Krebs, E. E., & Monahan, P. O. (2020). Minimally important differences and severity thresholds are estimated for the PROMIS depression scales from three randomized clinical trials. J Affect Disord, 266, 100-108. https://doi.org/10.1016/j.jad.2020.01.101

Leung, J., & Li, K. K. (2023). Faith-Based Spiritual Intervention for Persons with Depression: Preliminary Evidence from a Pilot Study [Article]. Healthcare (Switzerland), 11(15), Article 2134. https://doi.org/10.3390/healthcare11152134

Li, W., Denson, L. A., & Dorstyn, D. S. (2018). Understanding Australian university students’ mental health help‐seeking: An empirical and theoretical investigation. Australian Journal of Psychology, 70(1), 30-40. https://doi.org/10.1111/ajpy.12157

Löwe, B., Unützer, J., Callahan, C. M., Perkins, A. J., & Kroenke, K. (2004). Monitoring Depression Treatment Outcomes with the Patient Health Questionnaire-9. Medical care, 42(12), 1194-1201. https://doi.org/10.1097/00005650-200412000-00006

Nasution, L. A., Afiyanti, Y., & Kurniawati, W. (2020). Effectiveness of spiritual intervention toward coping and spiritual well-being on patients with gynecological cancer. Asia-Pacific journal of oncology nursing, 7(3), 273-279.

Oakley, L., Kinmond, K., & Blundell, P. (2024). Responding well to Spiritual Abuse: practice implications for counselling and psychotherapy. British Journal of Guidance & Counselling, 52(2), 189-206.

Portnoff, L., McClintock, C., Lau, E., Choi, S., & Miller, L. (2017). Spirituality Cuts in Half the Relative Risk for Depression: Findings From the United States, China, and India. Spirituality in Clinical Practice, 4(1), 22-31. https://doi.org/10.1037/scp0000127

Post, L., Ganzevoort, R. R., & Verdonck-de Leeuw, I. M. (2020). Transcending the Suffering in Cancer: Impact of a Spiritual Life Review Intervention on Spiritual Re-Evaluation, Spiritual Growth and Psycho-Spiritual Wellbeing. Religions (Basel, Switzerland ), 11(3), 142. https://doi.org/10.3390/rel11030142

Taylor, E. H. (2023). The myth of spirituality. Journal of Social Work, 23(6), 1005-1021.

Wong, V. W.-H., Ho, F. Y.-Y., Shi, N.-K., Tong, J. T.-Y., Chung, K.-F., Yeung, W.-F., Ng, C. H., Oliver, G., & Sarris, J. (2021). Smartphone-delivered multicomponent lifestyle medicine intervention for depressive symptoms: A randomized controlled trial. Journal of Consulting and Clinical Psychology, 89(12), 970.

_________________________

Appendix I

     Table S1. Concept mapping on integration of treatment strategies

Session

Content

Integration with CBT, positive psychology, and spirituality/religiosity

1

Spirituality, mental health, and depression

-Introducing cognitive reframing techniques to explore personal meaning of spirituality.

-Implementing positive strategies to introduce the concept of mental health and depression.

-Incorporating spiritual and religious beliefs to provide a foundation for understanding mental health and spirituality.

2

Connectedness

- Utilizing CBT techniques to challenge and reframe thoughts about connection with oneself and others.

-Applying positive psychology strategies to enhance feelings of connectedness and belonging.

-Incorporating spiritual practices and group rituals to strengthen connections and promote spiritual connectedness.

3

Forgiveness and freedom

-Implementing CBT exercises to address cognitive distortions related to forgiveness and freedom, while addressing individual uniqueness.

-Incorporating positive psychology interventions to explore steps towards forgiveness and promote emotional liberation.

-Integrating spiritual teachings on forgiveness and freedom to provide a framework for healing and reconciliation, and stronger interpersonal connection.

4

Suffering and transcendence

- Using CBT to reframe beliefs about suffering and develop coping strategies.

-Integrating positive psychology strategies to promote resilience, courage, and strength during times of suffering.

-Incorporating spiritual perspectives on suffering and transcendence to find deeper meaning and purpose in difficult experiences. Encourage shared experiences and discussions that promote mutual support and spiritual growth.

5

Hope

-CBT technique to challenge pessimistic thoughts and promoting optimistic.

-Integrating positive psychology interventions to foster a sense of optimism and hope.

-Incorporating spiritual teachings on hope and faith to inspire resilience and perseverance in challenging times.  Foster a supportive environment that nurtures hope and encourages interpersonal connections.

6

Gratitude

- Incorporating CBT to address cognitive biases and enhance gratitude practice.

-Applying positive psychology principles to explore personal ways of expressing gratitude and enhance well-being.

- Incorporating spiritual practices of gratitude and thankfulness to cultivate a sense of spiritual well-being and connection. Sharing gratitude practices within the group to strength bonds and promote a sense of community.

7

Relapse prevention and spiritual growth

- Implementing CBT strategies to identify triggers and develop relapse prevention plans.

-Incorporating positive psychology strategies to develop coping mechanisms and promote spiritual growth in times of challenge

- Integrating spiritual practices and teachings for relapse prevention and spiritual resilience in the face of difficulties. Establishing ongoing support networks and spiritual practices to sustain growth and connection beyond the sessions.

8

Wrap up and celebration

-Implementing positive psychology interventions to celebrate participants' accomplishments and foster a sense of achievement.

- Integrating CBT to allow participants to reflect on their learning and emotional growth throughout the sessions.

- Incorporating spiritual rituals and communal celebrations to acknowledge progress, growth, and spiritual connections within the group. Cultivating a sense of community and shared achievement through group reflections and celebratory activities.

      Note: CBT: cognitive behavior therapy 

_________________________

Appendix II

Information Sheet on the Research Study

Research Title:

Effectiveness of Spiritual Intervention for Persons with Depression /depressive mood: A Randomized Controlled Trial

Investigator:

Ms. Leung Lai Fun Judy, PhD Candidate

Department of Social and Behavioural Sciences, City University of Hong Kong

Phone: 852 3970 8798

Supervisor:

Dr. Ben K.K. LI, Associate Professor

Department of Social and Behavioural Sciences, City University of Hong Kong

Phone: 852 3442 8261

Background and Significance of the Study

            Depression is a common and serious mental health issue, and it is a major contributor to global disability. Spirituality and mental health have been a concern in the society. Religion and spirituality may be seen as coping skills, which mitigate stressors, provide a better sense of well-being, and decrease the overall risk for depression.

            Through the study, a theoretical framework of spirituality and depression will be developed from the data. It will provide a better understanding on service users’ spiritual needs and information for health professionals to plan and design appropriate intervention for clients.

Aim of Research:  The aim of the study is to explore the effectiveness of spiritual intervention in persons with depression.

Invitation to Participate

You are invited to take part in this research. Before you decide to participate, it is important for you to understand why the research is being done and what it will involve. Please take time to read the following information and take time to decide whether you will take part in this research or not. You are encouraged to contact the researcher via the phone numbers or email addresses provided above if you would like further information regarding any aspect of this project.

Research Method and Process

            If you are willing to participate, you will be randomly assigned to either group A or group B in different time points. The programme involves 8 sessions, each session lasting for around two hours. Participants will learn different topics each week to cope with depression symptoms and participate in pre-scheduled activities such as singing hymns, scripture reading and prayers, etc. You will be invited to take part in a focus group interview of around one and half hours to collect information about your views and perceptions on the spiritual intervention programme. The training sessions and the focus group interview will be audio-taped for purpose of analysis. Your valuable opinion will help to explore the specific aspects of spirituality.

            As a participant, you will be invited to fill out questionnaires at three time points, i.e. before, after, and at three months follow-up. It will take you about 20 minutes to fill out the questionnaires each time.

Consenting to participate and withdrawal from the research

            You are asked to consider to volunteer to take part in this project. Consenting to take part in this study involves signing and returning the consent form. Participants have the right to withdraw from the study at any stage up until the stage when the data is entered onto an electronic database. There are no negative consequences for declining to take part in the study or deciding to opt out of the study. You have the right to withdraw from the study at any time with no consequence.

Possible benefits and risks to participants

            Benefits will include learning about the knowledge and skills to cope with your depression, and support from peers who have similar problems. Your participation certainly enhances the understanding of therapeutic effect of spiritual intervention. There is no known physical, social, emotional, economical or psychological risks to yourself in taking part in the study.

Confidentiality

            Participation in the study will be completely anonymous and individual respondents will not be identifiable. No identifiable information is being collected in the questionnaire so your anonymity is ensured. Result of the study will be used for the completion of a doctoral thesis. The research information will also be presented at professional conferences and will be published in peer-reviewed journal. Confidentiality of participants’ details and data will be ensured since no individual results will be reported, instead only aggregate de-identified data will be reported/published.

Storage of data

            Hard copies of the completed questionnaires will be stored in a locked filing cabinet in a secured office. Data will be entered onto an electronic database. This database will be stored in a USB with password protected. Hard copies of the questionnaires will be shredded and the electronic database will deleted after the completion of the study.

Questions and Concerns

            If have any concerns about the study, please feel free to contact the investigator Ms. Judy Leung at 39708798. If you have enquiry about the ethical approval issue and/or your rights as a participant of this research study, please contact the Human Subjects Ethics Sub-committee of City University of Hong Kong (email: [email protected], telephone number: 852 3442 6856) or the Research Ethics Committee of HKMU at 27686251.

_________________________

Appendix III

Informed Consent for Interview in English

City University of Hong Kong

Department of Social and Behavioural Sciences

CONSENT FORM

Project Title  : Effectiveness of Spiritual Intervention for Persons with Depression/Depressive Symptoms: A Randomized Controlled Trial.

Investigator  : LEUNG Lai Fun Judy, Student of PhD Social & Behavioural Sciences, City University of Hong Kong

As part of my study in the PhD, I am conducting a research project under the supervision of Dr. Ben K.K. Li, Associate Professor of Department of Social & Behavioural Sciences in City University of Hong Kong.  The purpose of this study is to explore the effectiveness of spiritual intervention in persons with depression.  The programme includes eight sessions, each session lasting for two hours. The intervention session will be conducted once per week. You will be invited to take part in a focus group interview around one and half hours in length to collect information about your views, feelings and experience after the programme.  The training sessions and the focus group interview will be audio-taped for purpose of analysis.  The final report will only contain anonymous quotations.  Your name will not be used in the study and you will not be identified in any way.  Data will be stored securely and will only be accessed by the researcher.

THIS IS TO CERTIFY THAT, I, ______________________ HEREBY agree to participate as a volunteer in the above-named project.

I hereby give consent to be take part in the programme and the focus group interview as specified above.

I also give permission to for the sessions and focus group interview to be video recorded.  I understand that, at the completion of the research, the tape will be erased, and all the data will be destroyed.

I also understand that I am free to withdraw my consent and terminate my participation at any time, without penalty.  The service that I received will not be affected.

I have been given the opportunity to ask whatever questions I desire, and all such questions have been answered to my satisfaction.

Signature of Participant ___________________

Name of Participant ______________________              

Signature of Researcher ___________________

Name of Researcher_______________________

Date ___________________________________

Reviewer 3 Report

Comments and Suggestions for Authors

This is a well-presented, thoughtful, and scientifically sound study. My only note is the following. The authors might consider the extent to which the intervening psychotherapy is best characterized as 'Christian' or even 'grounded in Christianity' (p.4) as opposed to eclectically spiritual and cognitive-behavioral.

There are a few dimensions to this question. One is just the simple issue of consistency and accuracy in description. Another is the extent to which the clinical trials provide evidence of the efficacy of Christian prayer, specifically, or merely evidence of the efficacy of one's own active participation is some form of spiritual exercise or treatment program. The authors' own statements are at times ambiguous in this respect. For example, the authors state: 'Prayer's interaction with a divine/higher power, combined with a belief in divine support, can impact feelings of self-worth' (p.14). Here the authors seem to claim that divine agency is a causal factor. However, for most of the  article, the authors seem to focus only on the subjects' cognitive and behavioral states as the target of interest.        

I see no reason why a similar clinical trial could not also provide evidence of the efficacy of prayer - and in particular the efficacy of supernatural agency - in producing health outcomes. But it seems to me that this study was not designed to discover this. So caution might be taken in reflecting on this in the discussion. 

The extent to which the content of specifically Christian doctrine provides constraint on the intervening therapy program has implications for how one would qualify recommendations of this kind of intervention to Christian churches, missions, and community organizations. 

The authors mention the use of hymns, bible verses, the cultivation of forgiveness, and belief in an afterlife as features of the program. However, in the contemporary world, such features can be - and have been -  easily detached from the core Gospel message and core Christian doctrines that are focused on the actions, identity, and accomplishments of Jesus. 12 Step groups and other treatment/recovery program, not to mention various indigenous spiritualities, have many of the ritual and cognitive features that have been incorporated into the present study. If the present study has set itself the goal of discerning the effectiveness of specifically Christian spiritual connectivity, it is not clear to me that it has done this.             

Author Response

We would like to express our sincere gratitude to reivewer for your support and positive comments on our manuscript. your feedbacks are so encouraging and have strengthen our confidence in the quality of our work. We appreciate the time and effort in reviewing our manuscript, and the kind words that motivate us to continue our research work. The detailed responses are summarized in 3. Point-by-point response to Comments and Suggestions for Authors.

Point-by-point response to Comments and Suggestions for Authors

Comments 1:  This is a well-presented, thoughtful, and scientifically sound study. My only note is the following. The authors might consider the extent to which the intervening psychotherapy is best characterized as 'Christian' or even 'grounded in Christianity' (p.4) as opposed to eclectically spiritual and cognitive-behavioral.

Response 1: Thank you for your insightful feedback on our study. We appreciate your attention to the characterization of the psychotherapy employed in our research.

Our study is indeed grounded in Christianity within a group therapy framework, where we integrate principles from cognitive-behavioral therapy and positive psychology intervention. The intervention aims to provide participants with a holistic approach that combines the caring relationships within the church family with therapeutic components such as altruism, universality, hope, cohesiveness, and catharsis.

In our study, we utilize strategies rooted in Christian faith, such as studying Bible verses, prayer, hymn singing, and cognitive reframing through spiritual practices. These activities serve as avenues for cognitive restructuring and emotional healing. Additionally, positive psychology intervention strategies and skills, including gratitude practices, are interwoven with Christian principles to enhance the well-being of participants.

We believe that our approach offers a unique blend of Christian-based support and evidence-based psychological interventions, creating a comprehensive and culturally sensitive therapeutic environment that aligns with the values and beliefs of the participants.

We will make sure to clarify these aspects in our study to provide a more thorough understanding of the intervention and its integration of Christian faith with therapeutic strategies.

Comments 2:  There are a few dimensions to this question. One is just the simple issue of consistency and accuracy in description. Another is the extent to which the clinical trials provide evidence of the efficacy of Christian prayer, specifically, or merely evidence of the efficacy of one's own active participation is some form of spiritual exercise or treatment program. The authors' own statements are at times ambiguous in this respect. For example, the authors state: 'Prayer's interaction with a divine/higher power, combined with a belief in divine support, can impact feelings of self-worth' (p.14). Here the authors seem to claim that divine agency is a causal factor. However, for most of the  article, the authors seem to focus only on the subjects' cognitive and behavioral states as the target of interest.  

Response 2:  We greatly appreciate your thoughtful input on the manuscript. Your comments regarding the importance of consistency and accuracy in descriptions, as well as the distinction between the efficacy of Christian prayer and general spiritual exercises, are invaluable. Prayer is very powerful indeed, but it is just one of the components of the spiritual connectivity intervention programme.

While we acknowledge the need for clarity in defining the intervention's nature and mechanisms, we will keep your suggestions in mind for future research projects to ensure precision in delineating these aspects. Your observations on the focus of the study on cognitive and behavioral states are also duly noted, and we will strive to integrate discussions on the interplay between these dimensions and spiritual influences in our future work. Your thoughtful insights will undoubtedly contribute to the refinement of our research endeavors. Thank you for your time and valuable input.

Comments 3: I see no reason why a similar clinical trial could not also provide evidence of the efficacy of prayer - and in particular the efficacy of supernatural agency - in producing health outcomes. But it seems to me that this study was not designed to discover this. So caution might be taken in reflecting on this in the discussion. 

Response 3: We appreciate your thoughtful insights on the manuscript. Your point regarding the potential efficacy of prayer and supernatural agency in producing health outcomes is well taken. While our study was not specifically designed to investigate this aspect, we acknowledge the value of considering the role of prayer in health interventions.

We are familiar with the existing literature on the efficacy of prayer, as highlighted by studies such as Boelens et al. (2012); De Aguiar et al. (2017); Johnson (2018); Wachholtz and Sambamthoori (2013). In our study, we aimed to offer a comprehensive program that incorporates various effective delivery activities beyond prayer alone. These activities include hymn singing, bible scripture, meditation, and practices promoting love, peace, and joy. By embracing a diverse range of cultural and religious backgrounds among participants, our intervention seeks to cater to the needs of a broad spectrum of individuals.

While we acknowledge the potential benefits of prayer and supernatural agency, our holistic approach aims to create an inclusive and welcoming environment where participants from different backgrounds can engage comfortably. By offering a multifaceted program, we strive to provide a nuanced and adaptable intervention that resonates with diverse populations.

Your feedback underscores the importance of balancing the discussion on prayer with a broader perspective on spiritual practices, and we will ensure that future research endeavors reflect this comprehensive approach. Thank you for your valuable commentary, which enriches the dialogue on the efficacy of spiritual interventions in mental health research.

Comments 4: The extent to which the content of specifically Christian doctrine provides constraint on the intervening therapy program has implications for how one would qualify recommendations of this kind of intervention to Christian churches, missions, and community organizations. 

Response 4: We appreciate the insightful observation regarding the potential influence of Christian doctrine on our therapy program. While our intervention incorporates elements of spirituality and religiosity, including themes that may resonate with Christian beliefs, it is important to note that the program is designed to be inclusive and adaptable across diverse belief systems. Our aim was to create an intervention that draws from universal principles of hope, forgiveness, gratitude, and self-reflection, which are core to many spiritual and religious traditions, including Christianity.

In tailoring recommendations for Christian churches, missions, and community organizations, we emphasize the universal values and psychological principles that underpin the intervention. By focusing on these shared elements, we believe that the program can be effectively implemented within Christian communities while respecting and aligning with their specific doctrines and values. Moreover, we are open to collaborating with Christian leaders and organizations to further refine and tailor the intervention to ensure its compatibility with Christian beliefs and practices, thus maximizing its effectiveness and relevance within these settings.

Comments 5:  The authors mention the use of hymns, bible verses, the cultivation of forgiveness, and belief in an afterlife as features of the program. However, in the contemporary world, such features can be - and have been -  easily detached from the core Gospel message and core Christian doctrines that are focused on the actions, identity, and accomplishments of Jesus. 12 Step groups and other treatment/recovery program, not to mention various indigenous spiritualities, have many of the ritual and cognitive features that have been incorporated into the present study. If the present study has set itself the goal of discerning the effectiveness of specifically Christian spiritual connectivity, it is not clear to me that it has done this.       

Response 5: We appreciate the reviewer's insightful comments regarding the spiritual components of our study. Our intention in incorporating hymns, bible verses, forgiveness cultivation, and belief in an afterlife was not to exclusively focus on a narrow interpretation of Christian spirituality, but rather to create a holistic and inclusive program that respects and accommodates individuals of all faiths and belief systems.

The spiritual elements included in our program were selected to promote a sense of connection, hope, and healing among participants, drawing on universal themes of compassion, forgiveness, and transcendence that resonate across diverse spiritual traditions. We aimed to create a safe and welcoming space where individuals with varying spiritual beliefs and cultural practices could engage with the program in a manner that was meaningful and relevant to them.

It is important to emphasize that our study is not narrowly focused on discerning the effectiveness of specifically Christian spiritual connectivity. Instead, our approach integrates elements of spirituality and religiosity within a broader theoretical framework that combines group therapy, cognitive behavioral therapy, and positive psychology interventions. By blending these approaches, we aimed to offer a comprehensive program that addresses the multifaceted needs of participants while respecting their individual spiritual journeys and cultural backgrounds.

Throughout the intervention, participants with diverse spiritual beliefs were treated with respect and sensitivity, ensuring that the program's spiritual components were inclusive and adaptable to a range of perspectives. We recognize the importance of creating a supportive and non-judgmental environment where individuals feel empowered to explore and incorporate spiritual practices that align with their personal values and experiences. In fact, many participants were open and willing to share their experience of salvation through faith, repentance, and forgiveness, and how they cope with the mental health problems through prayer and support from church friends. Their genuine sharing can be beneficial to other participants as it also enhance hope in the journey of recovery.

We value the reviewer's feedback and will continue to refine our program to ensure that it remains inclusive, culturally sensitive, and responsive to the diverse spiritual needs of our participants. Thank you for highlighting these considerations, which will further enrich the impact and effectiveness of our intervention.

We hope the above points can clarify our intentions for the study. Thank you again for your feedback and for taking time to review my work.

References:

Boelens, P. A., Reeves, R. R., Replogle, W. H., & Koenig, H. G. (2012). The Effect of Prayer on Depression and Anxiety: Maintenance of Positive Influence One Year after Prayer Intervention. The International Journal of Psychiatry in Medicine, 43(1), 85-98. https://doi.org/10.2190/PM.43.1.f

De Aguiar, P., Tatton-Ramos, T., & Alminhana, L. (2017). Research on Intercessory Prayer: Theoretical and Methodological Considerations. J Relig Health, 56(6), 1930-1936. https://doi.org/10.1007/s10943-015-0172-9

Johnson, K. (2018). Prayer: A Helpful Aid in Recovery from Depression. J Relig Health, 57(6), 2290-2300. https://doi.org/10.1007/s10943-018-0564-8

Wachholtz, A., & Sambamthoori, U. (2013). National Trends in Prayer Use as a Coping Mechanism for Depression: Changes from 2002 to 2007. J Relig Health, 52(4), 1356-1368. https://doi.org/10.1007/s10943-012-9649-y

Reviewer 4 Report

Comments and Suggestions for Authors

I sincerely thank the authors on presenting such a robust and rigorous study. I know this work offers a great contribution to an emerging area in terms of the impact of spirituality in improving mental health. This is also a key area of mine that I am conducting in the Australian context with Māori and Pacific peoples given the significant role spirituality (Christianity) plays within our culture and community. I offer these studies as a possible suggestion to your paper if it will be of use. 

Fainga'a-Manu Sione, I., Faleolo, R. & Hafu-Fetokai, C. (2024). Finding Harmony between Decolonization and Christianity in Academia. Art/Research International, 8(2), 519–546. https://doi.org/10.18432/ari29764

Author Response

We would like to express our sincere gratitude to reivewer for your support and positive comments on our manuscript. your feedbacks are so encouraging and have strengthen our confidence in the quality of our work. We appreciate the time and effort in reviewing our manuscript, and the kind words that motivate us to continue our research work. The respoonses are summarized in point-by-point response to Comments and Suggestions for Authors.

Point-by-point response to Comments and Suggestions for Authors

Comments 1:  I sincerely thank the authors on presenting such a robust and rigorous study. I know this work offers a great contribution to an emerging area in terms of the impact of spirituality in improving mental health. This is also a key area of mine that I am conducting in the Australian context with Māori and Pacific peoples given the significant role spirituality (Christianity) plays within our culture and community. I offer these studies as a possible suggestion to your paper if it will be of use. 

Fainga'a-Manu Sione, I., Faleolo, R. & Hafu-Fetokai, C. (2024). Finding Harmony between Decolonization and Christianity in Academia. Art/Research International, 8(2), 519–546. https://doi.org/10.18432/ari29764

Response 1: Thank you for your thoughtful and encouraging comments on our article.  Your feedback means a lot to us, and we are grateful for your positive assessment of our work.

We also want to express our gratitude for sharing your article, "Finding Harmony between Decolonization and Christianity in Academia," with us. It is insightful and thought-provoking to read about your exploration of decolonization and Christianity within academia, especially from the perspective of Oceanian women.

Your recognition of the importance of spirituality in improving mental health, particularly within the Australian context with Māori and Pacific peoples, aligns closely with the core themes of our research. We believe that our studies complement each other well and contribute positively to the ongoing dialogue in our respective fields.

Thank you once again for your support and for offering valuable suggestions for further reading. We look forward to the opportunity of exploring the resources you have recommended and continuing to engage in meaningful discussions on these critical topics.

Round 2

Reviewer 1 Report

Comments and Suggestions for Authors

Dear Authors, 

Thank you for all your edits and for clarifying my concerns. I am satisfied with the current version.

Reviewer 2 Report

Comments and Suggestions for Authors

acceptable in its current form.

Comments on the Quality of English Language

Minor editing of English language required